Manuscript prepared for Atmos. Chem. Phys.
with version 2014/09/16 7.15 Copernicus papers of the LaTeX class copernicus.cls.
Date: 5 January 2017

# Chemistry-climate interactions of aerosol nitrate from lightning

Tost H.[1]

[1]Institute for Atmospheric Physics, Johannes Gutenberg University Mainz, Germany

*Correspondence to:* H. Tost (tosth@uni-mainz.de)

**Abstract.** Lightning represents one of the dominant emission sources for $NO_x$ in the troposphere. The direct release of oxidised nitrogen in the upper troposphere does not only affect ozone formation, but also chemical and microphysical properties of aerosol particles in this region. This study investigates the direct impact of $LNO_x$ emissions on upper tropospheric nitrate using a global chemistry climate model. The simulation results show a substantial influence of the lightning emissions on the mixing ratios of nitrate aerosol in the upper troposphere of more than 50%. In addition to the impact on nitrate, lightning substantially affects the oxidising capacity of the atmosphere with substantial implications for gas phase sulphate formation and new particle formation in the upper troposphere. In conjunction with the condensation of nitrates, substantial differences in the aerosol size distribution occur in the upper troposphere as a consequence of lightning. This has implications for the extinction properties of the aerosol particles and for the cloud optical properties. While the extinction is generally slightly enhanced due to the $LNO_x$ emissions, the response of the clouds is ambiguous due to compensating effects in both liquid and ice clouds. Resulting shortwave flux perturbations are of $\sim -100\text{mW/m}^2$ as determined from several sensitivity scenarios, but an uncertainty range of almost 50% has to be defined due to the large internal variability of the system and the uncertainties in the multitude of involved processes. Despite the clear statistical significance of the influence of lightning on the nitrate concentrations, the robustness of the findings gradually decreases towards the determination of the radiative flux perturbations.

# 1 Introduction

Lightning is one of most energetic phenomenon in the Earth atmosphere. Due to the tremendous electricity, the associated temperatures allow for breaking up the stable molecular nitrogen compounds into fragments which partly recombine to nitrogen oxides (Schumann and Huntrieser, 2007). Hence lightning represents a natural atmospheric emission source for $NO_x$, in addition to the anthropogenic sources from industry, energy production, traffic and agriculture (e.g., Jaegle et al., 2005). Further-

more, biomass burning contributes significantly to the total $NO_x$ emissions (e.g., van der Werf et al., 2010). The global lightning $NO_x$ ($LNO_x$) production has been estimated to range between 2 and 8 Tg N/yr (Schumann and Huntrieser, 2007), hence being in a similar order of magnitude as the soil emissions (e.g., Steinkamp and Lawrence, 2011), which represent an additional important natural contribution to the total oxidised nitrogen in the atmosphere. In contrast to the other sources

(with the exception of aircraft emissions), $LNO_x$ represents an upper tropospheric source. Due to a different chemical composition and chemical reactivity compared to the boundary layer, chemical conversion into $HNO_3$ is relatively efficient and most important is not subject to fast removal by dry deposition. Once nitric acid is formed this can condense on existing aerosol particles, mostly thermodynamically stabilised by ammonium ($NH_4^+$), forming ammoniumnitrate ($NH_4NO_3$). This

aerosol species is considered semi-volatile such that in the lowermost atmosphere a substantial part of nitrate re-evaporates (e.g., Stelson et al., 1979). However, due to the lower temperatures in the upper troposphere, $NH_4NO_3$ is thermally stabilised, and remains mostly in the aerosol phase. The total amount of nitrate aerosol does not only depend on the available nitric acid, but also neutralising cations, i.e. mostly ammonium.Furthermore, the condensation rate of nitric acid forming nitrates also

depends on the available aerosol concentration, as well as competition for neutralising compounds with e.g. sulphate ions.

Experimental evidence for the occurrence of nitrate in convective outflows have recently been observed over the Amazon by aircraft measurements [1]. Furthermore, $NH_3$ has been observed based on satellite retrievals in the outflow of the Asian monsoon (Höpfner et al., 2016), leading to the conclu-

sion that not all soluble compounds are completely removed by scavenging during convective lifting. Consequently, there is observational evidence for both $NH_3$ and $NO_3^-$, such that the formation of $NH_4NO_3$ is likely due to the low temperatures in the upper troposphere. A recent study by Yang et al. (2015) also shows that the observed scavenging efficiencies for nitrate (together with $HNO_3$) and ammonium are around 80% for the DC3 campaign (Barth et al., 2015), which allows the con-

clusion that the observed $NO_3^-$ is most likely formed above the levels of substantial wet removal, e.g. by conversion of lightning $NO_x$.

Being a component of the mixture of aerosol particles, the nitrate aerosol from lightning can also influence atmospheric radiation via the direct and indirect aerosol effects by a multitude of pathways.

---

[1]J. Schneider, pers. comm., 2016

First, nitrate contributes to water uptake increasing ambient aerosol size at a given relative humidity. Both the particulate nitrate mass as well as the additional aerosol water enhance the radiative extinction properties of the particle and can even alter the single scattering albedo influencing the direct aerosol effect of nitrate particles (e.g., Adams et al., 2001).

Additionally, the overall chemical composition of the aerosol is modified by nitrate from lightning such that the cloud condensation nuclei (CCN) or ice nuclei (IN) efficiency of the particles changes with implications for the indirect aerosol effects. Consequently, this indirect effect should be considered a competition between the formation of additional CCN (e.g., Makkonen et al., 2012) by the nitrate and a potential deactivation of IN for contact or deposition freezing by providing a hygroscopic coating. Immersion freezing is also affected via the concentration of the solutions, IN deactivation and freezing point modifications.

Currently, only a few global aerosol chemistry climate models can realistically simulate aerosol nitrate and its implications for the direct and partially also indirect aerosol effect due to the complex chemical interactions and the semivolatility (e.g., Jacobson, 2001; Adams et al., 2001; Bauer et al., 2007; Bellouin et al., 2011; Xu and Penner, 2012; Makkonen et al., 2012). However, regional scale air quality models often consider the formation of nitrate and also implications of the emissions from various sources on the $NO_3^-$ formation, e.g. Zare et al. (2014), who investigate the role of $LNO_x$ on PM2.5 in Europe or Allen et al. (2012), who among other aspects analysed the influence of lightning on nitrate wet deposition in the US. The effect of $LNO_x$ on gas phase chemistry and the oxidation capacity of the atmosphere has been previously studied and is continuously revisited (e.g., Labrador et al., 2004; Schumann and Huntrieser, 2007; Gressent et al., 2016; Finney et al., 2016a, b).

Usually, the effect of nitrates is determined from *annihilation studies*, i.e. the effect of nitrates is completely ignored. However, this might have implications for the overall aerosol chemical composition and size distribution such that these effects are mixed with the pure impacts of nitrate. Instead of a total nitrate *annihilation scenario*, this study investigates the omission of only a fraction of the aerosol nitrate, which originates from the lightning $NO_x$ emissions. According to our knowledge, the resulting climate impacts from lightning caused by particulate nitrate have not yet been quantified with a comprehensive chemistry climate model. Nevertheless, this scenario reveals already strong impacts due to the substantial implications of $LNO_x$ emission on tropospheric chemistry. However, emission sensitivity studies on $LNO_x$ would suffer from an even worse signal-to-noise ratio than found in the current study.

The results from this study are in agreement with previous studies as mentioned above with respect to the impact on ozone and the oxidation capacity, but explicitly analyse the impact of $LNO_x$ on upper tropospheric nitrate concentrations and its implications for the climate system.

## 2 Model description

### 2.1 The EMAC modelling system

This study applies the ECHAM5/MESSy Atmospheric Chemistry (EMAC) model, which is a numerical chemistry and climate simulation system that includes sub-models describing tropospheric and middle atmosphere processes and their interaction with oceans, land and human influences (Jöckel et al., 2010). It uses the second version of the Modular Earth Submodel System (MESSy2) (Jöckel et al., 2005) to link multi-institutional computer codes. The core atmospheric model is the 5th generation European Centre Hamburg general circulation model (ECHAM5, Roeckner et al., 2006). For the present study we applied EMAC (ECHAM5 version 5.3.02, MESSy version 2.50) in the T42L31-resolution, i.e. with a spherical truncation of T42 (corresponding to a quadratic Gaussian grid of approx. 2.8 by 2.8 degrees in latitude and longitude) with 31 vertical hybrid pressure levels up to 10 hPa. The applied model setup comprised the submodels for radiation, convection, large-scale clouds and condensation and the budget (source, transport and loss processes) of chemical compounds in the gas and aerosol phase.

### 2.2 Aerosol climate processes within EMAC

To simulate the relevant processes mentioned above and sketched in Fig. 1, we employed the lightning $NO_x$ emissions scheme by Price and Rind (1992). This scheme uses convective cloud top height as simulated by the convection scheme (Tiedtke, 1989). Even though such a combination of two parameterisations is subject to large uncertainties, we have shown in the past that this special combination is relatively robust and able to provide relatively realistic lightning distributions compared to satellite data from the LIS/OTD sensors (Tost et al., 2007). The emitted NO is subject to gas phase chemical transformations which are calculated with the help of the chemical model MECCA (Sander et al., 2011). The resulting nitric acid ($HNO_3$) can subsequently partition into the aerosol phase. The corresponding aerosol processes are simulated with the GMXE aerosol submodel (Pringle et al., 2010; Tost and Pringle, 2012) taking both the gas-aerosol phase partitioning and the interactions with other chemical compounds as well as the microphysical properties of the aerosol particles into account. The gas-aerosol phase partitioning of inorganic semivolatile compounds is calculated by the ISORROPIA2 model (Fountoukis and Nenes, 2007), which is part of the GMXE aerosol scheme. The aerosol particles are discretised in 4 lognormal size categories, and for the larger three modes a distinction is done between internally mixed hydrophilic particles and externally mixed hydrophobic particles resulting overall in 7 lognormal modes. Additional anthropogenic and natural emissions except for lightning are simulated with the submodels ONEMIS, OFFEMIS and TNUDGE (Kerkweg et al., 2006b) submodels, providing sources for other primary and secondary aerosol particles. Physical loss processes (dry and wet deposition, additionally sedimentation for aerosol processes)

are explicitly considered in the schemes DRYDEP, SCAV and SEDI (Kerkweg et al., 2006a; Tost et al., 2006).

To consider aerosol radiation interactions the prognostic aerosol mass and number concentrations (including aerosol water) are used in lookup tables from Mie calculations precalculated with the LIBRADTRAN (Mayer and Kylling, 2005) to determine the radiative properties of the atmospheric aerosol (extinction, single scattering albedo, asymmetry parameter) as described in detail in Pozzer et al. (2012); de Meij et al. (2012) and Dietmüller et al. (2016). These parameters are explicitly considered in the radiation scheme native to ECHAM5 (Roeckner et al., 2003) replacing the aerosol climatology by Tanre et al. (1984) as described in Dietmüller et al. (2016). For the treatment of indirect aerosol effects, we have implemented a two moment cloud microphysics scheme (Lohmann and Hoose, 2009; Lohmann et al., 2010). The activation of aerosols is calculated with the scheme of Abdul-Razzak and Ghan (2000), which has been adapted to the simulated aerosol types. Furthermore, the interactions of aerosols in the homogeneous and heterogeneous freezing processes (Kärcher et al., 2006) are considered, including an adaption to the more comprehensive chemical composition of the aerosol simulated with GMXE. To investigate the sensitivity of the climate impacts of aerosol particles influenced by lightning caused by aerosol cloud interactions (ACI), a second set of simulations has been performed using a modified activation scheme based on a combination of the work of Abdul-Razzak and Ghan (2000) and Petters and Kreidenweis (2007), as described in detail by Chang et al. (2014). The set of simulations using the Abdul-Razzak and Ghan (2000) aerosol activation scheme will afterwards be abbreviated with ARG, for the other set of simulations following Chang et al. (2014) the KK (i.e. kappa-koehler) acronym will be used.

Both schemes include the effects of nitrate, but different approaches are followed to calculate the critical supersaturation: the ARG scheme uses parameters for the osmotic coefficient and dissociation of nitrate in the solution (which are more uncertain than for $(NH_4)_2SO_4$), whereas the KK approach uses a volume weighted $\kappa$ to determine the total aerosol water uptake and hence cloud formation potential. The two approaches are therefore merely a different representation of the Raoult effect in the cloud activation. Note, that the aerosol-cloud interactions are only taken into account for large-scale clouds as the treatment of cloud microphysics in convective clouds is even further simplified.

## 2.3 Simulation Setup

Decadal simulations including all feedback mechanisms with the EMAC model have been performed for present day and preindustrial conditions. Sea surface temperatures are prescribed by a climatology from the AMIPII database for all model configurations. To determine the effects of aerosols caused by lightning two simulations are performed for each scenario, one with and one without $LNO_x$ emissions. Even though this annihilation scenario potentially cannot capture some compensation effects, we choose this approach due to the expected lower signal to noise ratio. Furthermore,

as there is nitrate produced from anthropogenic, biomass burning and soil sources of $NO_x$, there is no complete annihilation of all nitrate. For the present day scenario we applied emissions from the AC-CMIP emission inventory (Lamarque et al., 2010, 2013) for trace gases and aerosol emissions from the AEROCOM (Dentener et al., 2006) experiment, for preindustrial conditions we followed the AEROCOM (Dentener et al., 2006) recommendation for preindustrial conditions. Biomass burning

is included using the GFED data (van der Werf et al., 2010), as well as a compilation for preindustrial biomass burning. Note, that prescribed aerosol emissions for dust and sea salt have been applied, not including a potential feedback of a changed circulation (wind speed and wind patterns) on aerosol sources. The feedback mechanism are allowed to create an individual climate state without nudging the simulation results towards reanalysis data. The model simulations have been initialised with

results from previous experiments (see Jöckel et al. (2016)) eliminating spin-up effects and allowing a well initialised atmospheric composition state also for long-lived compounds such as $CH_4$. Due to the comprehensive feedback mechanisms we refrain from comparing individual years of the simulations, but focus on the decadal mean values and distributions.

## 3   Results

### 3.1   Lightning and associated emissions

To estimate the effects of lightning and associated emissions, we have analysed the distribution of $NO_x$ emissions from lightning. Fig. 2 depicts in a 3D visualisation the temporal mean emitted $LNO_x$ for the present day scenario.

The magnitude of the $LNO_x$ emissions is displayed as an isosurface of $1 \cdot 10^{-16}$ kg/(m$^3$ s). A

second isosurface of $3 \cdot 10^{-16}$ kg/(m$^3$ s), which is visible as a darker shading embedded in the first isosurface, shows that the dominant emissions take place in the upper part of the $LNO_x$ plume. This is a consequence of the fact that the vertical emissions redistribution follows a C-shape profile according to Pickering et al. (1998), leading to enhanced emissions in the upper section of the plume. Even though more recent studies (e.g., Ott et al., 2010) suggest a different vertical emission distri-

bution, past studies with the EMAC model have shown good agreement with observations of several campaigns using the C-shape profiles (Huntrieser et al., 2007; Tost et al., 2010).

The outer emission isosurface is colour coded with the total aerosol nitrate mixing ratio (in mol/mol), depicting the amount of aerosol nitrate present in the emission peaks. Furthermore, the gray shaded isosurface visualises a level of 0.1 ppb$_v$ of aerosol nitrate. This shows that high aerosol

nitrate concentrations are located in the lower troposphere, but also enhanced mixing ratios (>100 ppt$_v$) of particulate nitrate can be found in the upper troposphere, partially concurring with the $LNO_X$ emissions (compare the values of up to 100 ppt$_v$ as marked by the color coding on the emission plume surfaces).

The enhanced $NO_3^-$ mixing ratios in the lower troposphere can be explained by the anthropogenic
$NO_x$ emissions as well as sources for neutralising cations such as $NH_4^+$ close to the surface.

The coloured map at the bottom of Fig. 2 depicts the mean flash frequency per $km^2$ and minute, which is relatively well confined to the tropical continents and in reasonable agreement with observations from LIS/OTD (updated data following Christian et al. (2003)). Consequently, the $LNO_x$ emissions are co-located over the same regions.

The total $LNO_x$ emissions are 5.95 Tg N/yr with a $1\sigma$-variability of 0.03 Tg N/yr over the simulated decade in the present day scenario (ARG), for the KK scenario a total emission strength from lightning with $6.04 \pm 0.03$ Tg N/yr is simulated. Note that the seasonal variability is substantially larger. More than 80% of the emitted $LNO_x$ is placed above 500 hPa altitude in both scenarios. The difference between the two scenarios is a consequence of slightly different meteorological conditions
which are caused by the feedback effects of the aerosol and cloud properties.

### 3.2 Tropospheric nitrate from lightning

#### 3.2.1 Chemical budgets and distribution of oxidised nitrogen compounds

The emitted $NO_x$ from lightning along with other $NO_x$ molecules in the atmosphere forms $HNO_3$ mostly via the gas phase reaction of $NO_2 + OH \rightarrow HNO_3$ on typical time scales of hours to days.
Even though this reaction is much slower than the conversion between NO and $NO_2$, it represents an important end to the circular conversion of the $NO_x$ species.

The $LNO_x$ emissions for the present day scenario result in $\sim 40\%$ higher nitric acid mixing ratios in the troposphere, and even up to $\sim 61\%$ between 500 hPa and the tropopause, compared to simulations without lightning $NO_x$ emissions (see Tab. 1). Even though the enhancement effect for
$N_2O_5$ is stronger in relative numbers, the overall mixing ratios for $N_2O_5$ are substantially smaller (almost two orders of magnitude) such that dinitrogenpentoxide plays a minor role for the highly oxidised nitrogen compounds in the troposphere. PAN is found to see a reduction by 15% for the whole troposphere and 21% in the upper troposphere when no lightning emissions are considered, therefore being less affected by the $LNO_x$ emissions due to a potential limitation of VOCs for PAN
formation.

The nitric acid molecules can condense on pre-existing aerosol particles forming $NO_3^-$ ions in the deliquesced aerosol solution or solid $NH_4NO_3$ or $NaNO_3$ crystals. Tropospheric nitrate mixing ratios are typically a factor of two to three lower compared to $HNO_3$. Nevertheless, the changes in upper tropospheric $NO_3^-$ mixing ratios are $\sim 32\%$ globally averaged.
Fig. 3 shows isosurfaces of tropospheric nitrate concentration differences between the simulation with $LNO_x$ emissions to the case without this source. The gray shaded isosurface depicts a 30% difference, the blue isosurface 45% differences and the red enclosed area 60% enhanced $NO_3^-$ mixing ratios, with only statistically significant (based on a two sided t-test of annual mean data with a

significance level of 90%) data shown. Even though the maximum $NO_3^-$ absolute differences occur
down to the surface, the largest relative differences are apparent in the UT. Most of these differences are constrained to the tropics where the strongest $LNO_x$ emissions are prevalent. However, the maxima in the differences are not directly co-located with the emission maxima (i.e. Central Africa, Amazonia and the maritime continent, see Fig. 2), but generally further downwind. This is a consequence of atmospheric transport during the time required for oxidation of $NO_x$ to $HNO_3$ and subsequent partitioning into the aerosol phase.

To provide an estimate of the total amount of nitrate in the upper troposphere the back panel of Fig. 3 depicts the zonal mean $NO_3^-$ mixing ratio with values between 10 and 100 $ppt_v$ for the tropical middle and upper troposphere where the impact of the $LNO_x$ emissions is largest.

The mean upper tropospheric column burden (500 hPa to the tropopause) in $mg/m^2$ is depicted by the coloured panel at the bottom of the plot. The turquoise isolines on this panel depict the 20%, 40% and 60% differences in the UT column burden between the simulations with and without $LNO_x$ emissions. These are of course co-located with the regions of the isosurfaces.

The figure shows that the tropical upper troposphere is a region in which substantial amounts of particulate nitrate can be found. Both visualisations of the differences (UT column burden and mixing ratios) show that a large contribution of the upper tropospheric $NO_3^-$ originates from lightning emissions, but also that a sufficient amount of neutralising cations is available to stabilise the nitrate in the particulate phase. These findings are a direct consequence of the emission and conversion of $NO_x$ to N(V), but of course are also indirectly affected by changes in the oxidation capacity of the atmosphere and hence the $HNO_3$ formation.

The tropospheric budget of the highly oxidised N compounds is summarised in Tab. 1, including the relative importance of the $LNO_x$ emissions. For present day conditions, the particulate nitrate contribution to the total N(V) is $\sim 25\%$, whereas in the upper troposphere it is only $\sim 13\%$. These numbers have been determined from Tab. 1 by the contribution calculation of the individual compounds from the total load.

The neglect of the $LNO_x$ emissions leads to a shift in the contribution of the particulate phase to the total N(V). Under these conditions 33% of the N(V) is in the form of particulate $NO_3^-$ for the whole troposphere, whereas for the UT the fraction is higher than 21%. This is a consequence of the reduced available N(V), but a similar amount of neutralising cations, i.e. mostly $NH_4^+$.

The sensitivity simulations in the KK configuration result in almost identical values, (budget is shown in Tab. 1 of the supplement) depicting the fact that the description of cloud and cloud removal processes plays a minor role for the total budget of particulate nitrate in both model configurations.

The budget of the loss processes of N(V) compounds can be found in the supplement for both present day and preindustrial conditions. The differences in the nitrate burden are similarly present in the loss processes, with wet deposition being mostly influenced (50% change in the loss flux), but sedimentation and dry deposition also equally contribute to the change in the total loss.

### 3.2.2 Influences on other chemical species

The reduced $NO_x$ burden in the simulation without lightning emissions has substantial impact on the concentrations of other species. As only a minor fraction of the total $LNO_X$ emissions of $\sim 6$ Tg N/yr are converted to N(V+) (e.g. 186 Gg N change in N(V) according to Tab. 1), the impact on ozone is comparable to previous studies (e.g., Labrador et al., 2005; Finney et al., 2016a): the calculated values for tropospheric ozone are 362 Tg in the simulation without lightning and 444 Tg if lightning is included, which corresponds to a 22% increase in the tropospheric ozone burden (even though the absolute values for the tropospheric ozone burden differ substantially in each of the studies). The effect of the $LNO_x$ emissions is again very prominent in the upper troposphere, where the $O_3$ load increases from 195 to 248 Tg by $LNO_x$ emissions, i.e. an increase of 27%. Driven by the location of most of the emissions, these effects mostly occur in the low latitudes. The resulting decrease in ozone in the zonal average resulting from neglecting lightning emissions is comparable to the findings of Grewe (2007). The change in the column burdens in the tropics also agrees with the results of Martin et al. (2002), both in the geographic patterns and the amplitude of the signal. Therefore concluding, the explicit consideration of the nitrate formation has no important impact on the tropospheric ozone distribution, such that the results of impact studies of lightning $NO_x$ (such as e.g., Banerjee et al., 2014) do not have to be revised.

However, the OH concentrations and therefore the oxidation capacity of the atmosphere are also substantially influenced by lightning (Labrador et al., 2004) via changes in $O_3$ and subsequent OH production via ozone photolysis and reaction of the products with water vapour. The effect on OH is displayed by a modification of the methane lifetime to additionally take the OH recycling capacity into account, as depicted in Tab. 2. The emissions of $LNO_x$ are responsible for an increase of the tropospheric methane lifetime of $\sim 1.7$ to 1.9 years; however in the upper troposphere (above 500 hPa up to the tropopause) an increase of the $CH_4$ lifetime of almost 10 and 13.2 years for present and preindustrial conditions, respectively, corresponding to almost a halving of the oxidation capacity of the upper troposphere. The direct OH concentrations differ only by $\sim 10$ to 20% showing the importance of the recycling mechanisms on the oxidation capacity of the atmosphere.

This is especially relevant for the production of sulphate via the gas phase oxidation of $SO_2$, which substantially influences aerosol formation and aerosol composition in the upper troposphere. The sulphate burden for present day conditions decreases from 576 Gg S(VI) to 565 Gg. This is also prominent in the UT region, where instead of 74.6 Gg S(VI) only 70.2 Gg S(VI) are simulated. As most of the oxidation in the upper troposphere takes place in the gas phase, the aqueous phase oxidation in the lower troposphere is affected to a minor degree. Additionally, most of the emitted $SO_2$ in the lower troposphere originates from anthropogenic sources, where also the co-emission of $NO_x$ is prevalent such that the oxidant levels for aqueous phase oxidation of S(IV) to S(VI) are less affected.

Particulate ammonium mixing ratios are only affected to a minor degree by the $LNO_x$ emissions, too: for present day conditions the $NH_4^+$ burden slightly decreases without lightning emissions, but increases during preindustrial times. However, the changes in the atmospheric burden are lower than

305 $\pm 5\%$. These changes can be explained by the of lower sulphate and nitrate burdens and effects of the oxidation capacity on gaseous ammonium concentrations. A significant change due to $LNO_x$ emissions in the sulphate to bisulphate ratio is not simulated by the model despite the capabilities of the thermodynamic equilibrium model.

The KK simulations show a very similar behaviour in the burdens, the $CH_4$ lifetime and the

310 changes induced by the omission of the $LNO_x$ emissions, such that a detailed discussion is skipped here.

### 3.3 Aerosol microphysical effects

The influences of the chemical composition in the upper troposphere due to lightning are also reflected in the microphysical properties of the aerosol, which are described by the size distributions

at various locations. The regions in the tropics are mostly characteristic for the tropical continents, as the oceanic aerosol distributions usually contain substantially lower particle concentrations such that the distribution of those regions is dominated by the continental grid cells. The regions have been selected following the analysis of the significant changes in Sect. 3.2.1, but also to represent various conditions of typical aerosol size distributions where lightning can play an important or only

a minor role.

The temporal mean size distributions of the aerosol particles for various regions (the respective regions are mentioned above the individual panels) are shown in Fig. 4 as a 2D plot with the horizontal axis depicting the ambient aerosol diameter and the vertical axis the pressure altitude; the colour coding displays the relative difference in aerosol number concentration [in %] between the

325 simulation without and with lightning (NOLNOX - LNOX/LNOX). Furthermore, the absolute values of the size distribution [in $cm^{-3}$ in a logarithmic scale] are depicted by the contour lines with the solid lines representing concentrations larger than 1 per $cm^3$. The figure shows the simulation results for the present day conditions (ARG case). The hatched regions in the 2D size distribution diagram mark those areas in which the changes do not exhibit statistical significance.

Over the tropical continents (middle row in Fig. 4) in the upper troposphere enhanced nucleation mode particle numbers are simulated if lightning emissions are considered, which is a result of the enhanced S(VI) concentrations and consequently enhanced new particle formation. At approximately 10 nm diameter the particle number is smaller in case of $LNO_x$ emissions, due to the interaction of enhanced coagulation and condensation on the enhanced particle concentrations.

Also additional condensation on existing particles and enhanced coagulation with the small particles causes higher values of particle numbers at $\sim$ 80 nm in the LNOX case. For larger particles slightly enhanced values are found in case $LNO_x$ emissions are considered as a consequence of the more

efficient coagulation (due to higher particle numbers in the small mode) versus the slower condensational growth. For the lower troposphere the picture is more ambiguous and most changes are not statistically significant: in South America and Indonesia reduced nucleation takes place. However, the absolute particle concentrations are so low that this process is almost negligible, whereas in Central Africa enhanced new particle formation is caused by lightning emissions. For the larger particles the impact is rather small.

In the mid-latitudes (upper row in Fig. 4) relevant changes in the size distributions are only found in the Eastern US and China but little impact in Europe and Siberia (lower right panel). The first two regions are located slightly more southwards such that lightning frequencies are enhanced compared to the latter two regions (see bottom map of Fig. 2). The general pattern of the changes is comparable to the tropical continents.

In the Southern Atlantic region the changes in the size distribution profile are relatively small, as lightning emissions play a minor role for the total particle concentration as well as for the oxidation capacity of the atmosphere and therefore sulphate formation, but they are still robust. In the Central Pacific where enhanced nitrate concentrations are simulated in the upper troposphere (see Fig. 3), the changes in the aerosol size distribution are moderate with strongest signals in the middle troposphere (between 700 and 500 hPa). As lightning $NO_x$ emissions are substantially smaller over the ocean, the oxidation capacity is affected to a minor degree; however nitrate and gaseous N(V) are transported downwind from the source region affecting the nitrate concentrations and particle numbers, mostly via coagulation.

To analyse the impact of the $LNO_x$ emissions on the aerosol water uptake, the mean growth factor (GF = ambient diameter / dry diameter) is compared. Using this parameter has the advantage that, e.g. in contrast to the aerosol water content, it is independent of the aerosol size and particle number concentrations. The overall impact of lightning on the GF is relatively small: for present day conditions the nucleation and aitken mode depict a slight increase in the upper troposphere (up to 2%), whereas in the accumulation and coarse mode a decrease of up to $-2\%$ is simulated caused by $LNO_x$ emissions. Due to the relatively dry conditions in the upper troposphere, where the nitrate changes are strongest, the effect on water uptake is expected to be that small. Closer to the surface the differences in the growth factor are even smaller and ambiguous in their sign, lacking statistical significance.

### 3.4 Impacts on climate

### 3.4.1 Aerosol optical properties

As aerosol particles scatter and absorb solar and infrared radiation, the impact of the $LNO_x$ emissions is investigated with the help of Fig. 5 for the ARG present day simulation. The left panel shows both the mean column AOD at 550 nm wavelength (on the floor level) as well as the zonal

mean extinction (rear panel). The ceiling panel depicts statistical significant changes in the column AOD and the front panel visualises statistical significant changes in the extinction. Note, that for determining the zonal mean of statistical significant changes, only the individual data points which exhibit significance are selected and the average over these (selected) points is taken.

The rear panel depicts the zonal mean of the average extinction per km. Substantial extinction is simulated in the lower troposphere (below 700 hPa) with an additional enhancement of the optical depth between $20°$ and $50°$N. This is a consequence of both the distribution of water vapour and hence aerosol water supporting aerosol growth and extinction in the lower troposphere of the tropics, on the other hand driven by natural sources of dust (and associated extinction) as well as anthropogenic pollution (especially from East Asia). The relative changes of aerosol extinction due to $LNO_x$ are depicted in the front panel. The most substantial enhancement of aerosol extinction is simulated in the middle to upper troposphere in the tropics with zonal mean enhancements higher than $20\%$. In the uppermost troposphere reductions in extinction are also simulated as a consequence of neglecting the lightning emissions. Compared to the nitrate enhancements as analysed from Fig. 3, the maximum enhancement of aerosol extinction by $LNO_x$ is located further downwards in the middle troposphere, i.e. between 400 and 600 hPa, whereas the strongest nitrate enhancement has been simulated between 200 and 400 hPa. The reason for this downward shift is discussed further below.

The column AOD (floor panel) shows maximum values in the dust emission regions (e.g. Northern Africa) and the anthropogenic pollution centers (e.g. Eastern China). The climatological global mean value with present day emissions is 0.122, which is close to the observations as derived from MODIS (e.g., Mao et al., 2014). The influence of the $LNO_x$ emissions can be analysed with the help of the ceiling map in Fig. 5, which shows the percentage fractions of the changes in the simulated AOD. Globally a slight reduction to 0.121 can be found ($\sim 1\%$). However, in some regions even higher extinction is simulated in case of no nitrate formation from lightning, which is in contrast to the findings of Sect. 3.2.1, where no regions with enhanced nitrate have been simulated. However, only in very few regions a statistical significant signal in the column AOD can be found. This is a consequence of the fact that significant changes in extinction are mostly located in the upper troposphere which contributes less to total column AOD (cf. rear panel). The right panel of Fig. 5 depicts again the absolute values of AOD (floor panel) and extinction (rear panel), but also shows isosurfaces of the location of the regions of enhancement and reduction of extinction. The pale red depicts the $+10\%$ isosurface, whereas the embedded dark red regions mark an increase of the extinction due to $LNO_x$ of more than $+20\%$. On the other hand the pale blue marks regions of a medium reduction ($-10\%$) and dark blue regions (only at the tropopause) depict extinction reductions of more than $20\%$. The main enhancement of extinction by lightning emissions occur between 400 and 600 hPa over the tropical continents and slightly further downwind. This general pattern corresponds to the enhanced nitrate mixing ratios (c.f. Fig. 3), but is more restricted to the continents. In the extra-tropics an even higher extinction is simulated if no $LNO_x$ emissions are considered. This is a consequence of the

secondary effects caused by the emissions, i.e. the influence on sulphur oxidation via the oxidation capacity of the atmosphere and changes in the size distribution. Comparing the pattern correlation between the changes in nitrate and extinction a relatively low value is found ($R \approx 0.3$). However, the changes in sulphate and extinction show a higher correlation value of $R \approx 0.4$. Analysing the PDF of the differences in AOD between the simulation with and without lightning emissions (see

supplement) reveals that for present day conditions larger deviations occur more often compared to the preindustrial cases. The PDFs are almost symmetric, but show a slight shift towards enhanced extinction in the simulations including $LNO_x$ emissions.

The sensitivity simulations with the alternative warm cloud activation scheme in general show a similar distribution of the aerosol extinction with the enhancement in the middle and upper tropical

troposphere (see supplement). The global mean column AOD for present day conditions is slightly lower (0.120) and the reduction in case of no $LNO_x$ emissions even further reduced ($\sim 0.5\%$). Strongest enhancement is found over the tropical continents, i.e. the regions with maximum lightning activity, but most of the nitrate is simulated in the upper troposphere, whereas also in this configuration the extinction is enhanced in the middle to upper troposphere. For preindustrial condi-

tions, the enhancement of the extinction by lightning is a bit stronger, but still small ($\sim 1\%$).

To analyse whether the contribution of scattering versus absorption is influenced by the $LNO_x$ emissions, the single scattering albedo is compared in both model simulations. The zonal mean value changes by less than $\sim +1\%$ in the free troposphere and stratosphere, and is reduced by a maximum of $\sim -4\%$ in the boundary layer when lightning emissions are considered. Consequently, a change

in the thermodynamic structure resulting from direct aerosol-radiation interactions is expected to be small.

### 3.4.2 Cloud properties

As cloud properties can be influenced indirectly by the $LNO_x$ emissions three parameters are analysed to provide the causes for aerosol-cloud-radiation interactions. The cloud cover may hint at

modifications of the cloud lifetime effect (Albrecht, 1989), whereas the combination of the liquid (ice) water content and the cloud droplet (ice crystal) numbers, which can also be expressed via the effective radii, can shed light onto the direct influence of the lightning emissions on the radiative properties of the clouds via the Twomey (1977) effect.

– Cloud cover:

The cloud and precipitation cover changes only to a minor degree between the simulations with and without lightning for the present day scenario (ARG case). In the tropics the $LNO_x$ emissions lead to a small increase in the mid troposphere cloud coverage of the Northern hemisphere, whereas the dipole pattern is compensated by reduced cloudiness in the Southern hemisphere. Apart from this, only the polar latitudes show some larger (but not statistical

significant) changes in cloud coverage. Consequently, the impact of the cloud lifetime effect
        is expected to be small.

        – Liquid / ice water content:
        The patterns for cloud water and cloud ice are characterised by local dipole changes, but there
        are no significant modifications of the distribution of cloud water and ice, respectively. Only
the tropical middle to upper troposphere of the Northern hemisphere shows an enhanced ice
        mixing ratio in the simulation with lightning emissions, which correlates to the increase in
        cloud coverage. However, the decrease in cloud coverage in the corresponding region in the
        Southern hemisphere is not accompanied by a substantial decrease in ice water. In the zonal
        average the increase in total water is less than $10\%$.

– Cloud droplet / ice crystal numbers:
        Compared to the other two parameters, some substantial changes in cloud droplet and es-
        pecially in ice crystal number are found as a consequence of the $LNO_x$ emissions. For the
        present day scenario, in the troposphere between 650 and 400 hPa an increase in cloud droplets
        is simulated in case of active lightning emissions in the tropics. Further North a decrease of
cloud droplets between 900 and 600 hPa is calculated. Both are of approximately the same
        magnitude both in absolute and in relative terms and are a consequence of more hydrophilic
        particles due to nitrate coating and a reduction of lower tropospheric CCN due to reduction in
        lifetime and faster conversion from hydrophobic to hydrophilic conditions. However, the sta-
        tistical significance of the changes is relatively low. For the ice crystal number the influence
is even stronger: in the tropical upper troposphere the $LNO_x$ emissions lead to an increase in
        the ice crystal number in a region where the highest ice crystal numbers are simulated. In the
        mid-latitudes a reduction of the ice crystal numbers is simulated in both hemispheres above
        400 hPa. Both modifications are of the order of more than $\pm 20\%$. Due to the coarse tempo-
        ral resolution of the model output it is not possible to attribute the differences to a change in
regime from homogeneous to heterogeneous freezing. Nevertheless, it is possible that locally
        this effect can occur and has a substantial influence on the ice crystal number concentration.

By analysing the effective radii (Fig. 6), some of the effects on water mixing ratios and numbers
compensate, whereas others positively amplify each other. This is both valid for liquid clouds in
the lower and middle troposphere as well as ice clouds in the middle and upper troposphere. For
liquid clouds, the effective radii increase in the simulation with lightning in the polewards of $40°$
below 800 hPa and in the tropics ($0°$ to $20°$N) up to 400 hPa. However, most of the changes are not
statistically significant, such that Fig. 6 depicts only the changes in ice crystal size; a corresponding
figure for the liquid clouds can be found in the supplement. The back panel depicts in colours the
statistical significant zonal mean change in effective ice crystal size (in absolute units of $\mu$m). The
480 white contours represent absolute ice crystal sizes with maximum values in the upper part of the mid-

latitude storm tracks and the upper tropical troposphere. Note, that the absolute values (not marked) are of secondary importance as mean values over cloudy and non-cloudy conditions are taken, but the difference is robust. For ice clouds the effective radii also increase in the same tropical latitude range between 600 and 200 hPa. Additionally, a decrease in effective radii of similar magnitude on the other side of the ITCZ is simulated. Also in the high latitudes the effective ice crystal size mostly increases in the simulation with $LNO_x$ emissions, both in the upper troposphere as well as in the boundary layer. The shape of the regions of ice crystal changes extends vertically over more than 100 hPa and is not located in the uppermost troposphere only. This indicates that the statistical significant changes are more associated to the process of immersion freezing in mixed phase clouds than to the cirrus regime with competition between homogeneous and heterogeneous freezing.

Using the KK model configuration, which applies a different cloud droplet activation scheme, results in a lower sensitivity of the liquid droplet effective radius at elevated altitude for present day conditions. However, in the lower troposphere a similar strong signal is simulated, which leads to an increase of effective droplet radius in the tropical Southern hemisphere and a decrease in the Northern counterpart. For ice crystals, mostly a reduction of effective crystal size is simulated which is a consequence of the liquid droplet freezing. The effects for preindustrial conditions are similar, with even lower sensitivity of the low clouds on the lightning emissions.

In general, the regions with positive and negative effects in the effective droplet and crystal size show ambiguous signals, such that the impact on cloud radiative properties can only be addressed in conjunction with cloud coverage and the calculation of the radiative fluxes.

### 3.4.3 Radiative Fluxes

The combined effects of modifications in the aerosol extinction and the cloud optical properties lead to a modification of the shortwave and longwave radiation fluxes. In Fig. 7 the mean impact on the total sky shortwave flux is depicted in the upper panel. Shown are the differences in the SW flux at the top of the atmosphere between a simulation with and one without $LNO_x$ emissions (ARG configuration). Despite the relatively large internal variability of the system, several regions show a statistical significant change in the TOA short wave flux (based on a t-test with $90\%$ significance threshold on the annual mean data), which are marked by the hatched regions. However, the regions with increased or decreased fluxes are not directly co-located with the $LNO_x$ emissions. Down-wind of Central Africa an increased TOA short wave flux (corresponding to a heating) is simulated, whereas in Northern Amazonia and the maritime continent a decrease of the fluxes (corresponding to a cooling) is calculated. Also in the mid-latitude storm tracks mostly a slight increase in the fluxes is simulated, whereas over the tropical oceans the model suggests decreased fluxes. In the global mean the overall effect is determined as an enhanced backward reflection of the shortwave radiation of $\sim -0.1$ W/m$^2$ for all sky conditions. However, the time series of the global mean differences (depicted in the lower panel of Fig. 7) is characterised by large internal variability with

both positive and negative differences. The clear sky fluxes show a substantially weaker signal with only $\sim -0.05$ W/m$^2$ and hardly any regions with statistical significance. Furthermore, the patterns of substantial changes are clearly separated from the regions where lightning is dominant (i.e. mid-latitudes versus tropical continents). Consequently, the clear sky signal is interpreted as statistical noise.

Considering both the temporal variability of the mean and the large spatial variability, the uncertainty in the total effect has to be considered as relatively large. Using the sensitivity simulations with the alternative cloud activation scheme reveals for both present day and preindustrial conditions a shortwave flux perturbation of $\sim -0.06$ W/m$^2$ again with no significant contribution from the clear sky fluxes.

To bring these numbers into context with the total shortwave flux disturbances from anthropogenic aerosols, simulation results with both cloud activation schemes and present day and preindustrial emission scenarios are intercompared (for figures see supplement). Both model configurations show a statistical significant cooling over the regions of dominant anthropogenic aerosol pollution (Eastern US, Europe, China) for both clear sky and all sky conditions. The clear sky disturbance is $\sim -0.57$ for the ARG and $\sim -0.50$ W/m$^2$ for the KK configurations. All sky fluxes change by $\sim -1.62$ for the ARG and $\sim -1.42$ W/m$^2$ for the KK simulation setups, respectively. The overall anthropogenic aerosol effect is relatively large in the simulations compared to the results published in the latest IPCC report (IPCC, 2013) and therefore an overestimated sensitivity to the aerosol disturbance by lightning NO$_x$ nitrate cannot be ruled out completely.

The effect on the longwave radiation via both aerosol extinction and cloud effects is relatively small. Especially, the clear sky flux disturbances are almost negligible, such that the total effect is dominated by ACI.

Consequently, as the differences between both cloud activation schemes indicate an uncertainty of almost 50% the quantitative estimate of -100 mW/m$^2$ should also be used with a comparable uncertainty range of 50 mW/m$^2$.

## 4 Conclusions

The overall impact of chemically produced aerosol nitrate from lightning NO$_x$ emissions is analysed with a global chemistry climate model. Based on a total emission of 6 Tg N/yr, the contribution of LNO$_x$ to the concentrations of upper tropospheric oxidised nitrogen by more than 57% for present day and 75% for preindustrial conditions. Given a sufficient amount of neutralising cations, especially NH$_3$, substantial amounts of aerosol nitrate are formed with the increases in the upper tropospheric burden exceeding 30% for present day and 50% for preindustrial conditions. Therefore, lightning is also a major contributor to the aerosol nitrate burden in the upper troposphere.

The concentration enhancements are not uniformly distributed, but follow the regions of maximum emissions, and further downwind transport.

Besides the well known impacts of lightning on $O_3$, the impact on the chemical oxidation capacity of the atmosphere is highly important for the upper tropospheric aerosol loading. Because of a decrease of the oxidation potential (as represented by an increase in the methane lifetime), the sulphate formation via the gas phase reaction pathway is reduced in case of neglecting $LNO_x$ emissions. Hence new particle formation is suppressed, efficiently modifying the small part of the aerosol size distribution. In addition to the condensation of nitrate on the aerosol particles, the combination of a reduced particle number and the coagulation of the particles, substantial effects on the particle size distribution in the upper troposphere are simulated.

The changes in aerosol size, chemical composition and number concentrations have implications on climate via both aerosol-radiation and aerosol-cloud interactions. Lightning and its consequences (nitrate formation, size distribution changes) causes mostly an increase in the total aerosol extinction with pronounced increase maxima in the upper troposphere. On the other hand, both liquid and ice phase cloud optical properties are modified by the $LNO_x$ emissions, represented by e.g. effective droplet and ice crystal size. However, in contrast to the nitrate concentration increases the cloud effects resulting from lightning are ambiguous.

The resulting shortwave flux perturbations caused by $LNO_x$ emissions are quantified to be $\sim -100$ mW/m$^2$. The resulting effect is caused mostly by the aerosol-cloud interactions, whereas the direct aerosol-radiation interactions are of secondary importance. However, an uncertainty range of almost $\sim 50$ mW/m$^2$ has to be assumed due to large internal variability and uncertainties in the process description (mostly in the aerosol-cloud interactions).

Nevertheless, an increase in lightning activity in the future or impacts of a more efficient transformation of the $LNO_x$ emissions into nitrate under future climate conditions might have a non-negligible impact on the radiation balance of the atmosphere. On the other hand, aerosol nitrate formed from lightning offers further possibilities to address the feedback and (potentially compensating) impacts of combined chemistry-aerosol-climate interactions.

## Appendix A: Preindustrial conditions

The results from the sensitivity simulations for preindustrial conditions are summarised in this section.

- Emissions:

  In the preindustrial scenario total $LNO_x$ emissions of $6.12 \pm 0.03$ Tg N/yr are simulated for the ARG scenario and $6.16 \pm 0.03$ Tg N/yr for the KK simulations.

- Nitrate enhancement:

  During preindustrial times, an increase in the $HNO_3$ burden with $\sim 67\%$ and $\sim 76\%$ in the

total and upper troposphere is even stronger compared to present day conditions. This stronger enhancement is due to the larger contribution of the $LNO_x$ emissions compared to the total $NO_x$ release, which is mostly from anthropogenic sources in present day conditions. Enhancements for $N_2O_5$ are of $\sim 80\%$, but also under these conditions $N_2O_5$ is a minor contributor to the total highly oxidised nitrogen tropospheric load (less than $5\%$). A similar figure as Fig. 3 for preindustrial conditions can be found in the supplement. Due to the lower other sources of $NO_x$ in the atmosphere, the importance of the $LNO_X$ emissions is substantially increased. Therefore, the relative differences in $NO_3^-$ are larger which can be seen from the extended areas included by the respective isosurfaces and the larger areas covered by the turquoise contour lines at the floor panel of the figure. The lower overall nitrate mixing ratios are obvious from the colour scale of the column nitrate which is almost an order of magnitude lower compared to present day conditions. A lower contribution of particulate nitrate to the total N(V) load with $22\%$ for the whole and $9\%$ for the upper troposphere is simulated, which can be mostly attributed to lower $NH_3$ emissions and consequently less $NH_4^+$ ions to neutralise and thermodynamically stabilise available aerosol nitrate. However, the enhancement of the particulate phase due to $LNO_x$ is even stronger compared to present day conditions to $\sim 34\%$ for the whole troposphere and $17\%$ for the UT.

– Methane lifetime and sulphate:
The lifetime increase is even stronger without anthropogenic emissions, such that for the upper troposphere almost a doubling of the $CH_4$ lifetime occurs (see lower part of Tab. 2). Nevertheless, this results in a similar reduction of the sulphate burden compared to present day without $LNO_x$ emissions despite the substantially lower total $SO_2$ emissions.

– Size distributions:
For preindustrial conditions (see supplement) the simulation provides similar results from the tropics. However, in the mid-latitudes where a substantial reduction of other $NO_x$ emissions is applied compared to the present day scenario the impact of lightning on the nitrogen budget as well as the size distribution becomes more important. Obviously, a similar pattern as for the tropical continents is simulated; however, due to the weaker $LNO_x$ emissions in the mid-latitudes the effects are substantially smaller compared to the tropics.

– Growth factors:
The results for the GF are similar in their distribution compared to the present day scenario with a slightly increased amplitude (up to $\pm 5\%$). The sensitivity simulations *KK* are characterised by the same distribution and dependencies on the emissions by lightning. Corresponding figures can be found in the supplement.

– Extinction and AOD:
For preindustrial conditions the situation is quite comparable (see supplement), despite the

intensive AOD signals in the regions with anthropogenic pollution, especially East Asia. The simulated global mean column AOD at 550 nm is 0.090. Consequently, the enhancement of the extinction between $20°$ and $50°$N is less strong. The neglect of the lightning emissions results in a reduced column AOD in most regions, resulting in a global mean value of 0.089. A reduction in column AOD occurs only in regions of strong hydrophobic emissions (dust or BC) where the aerosol lifetime is reduced by a coating by nitrates and subsequent faster conversion from hydrophobic to hydrophilic categories. However, also in this case the statistical significance is relatively small. With respect to the zonal mean extinction the changes are slightly larger compared to present day conditions.

– Cloud response:

For preindustrial conditions the response to the $LNO_x$ emissions is comparable, but shows a slightly higher amplitude.

– Radiative fluxes:

For preindustrial conditions (for a graph see supplement) the situation is comparable with a slightly increased amplitude of $\sim -0.14$ W/m$^2$ and a similar pattern distribution. The clear sky signal is almost negligible with $\sim -0.01$ W/m$^2$ and no statistical significance.

*Acknowledgements.* Parts of this research were conducted using the supercomputer Mogon and advisory services offered by Johannes Gutenberg University Mainz (www.hpc.uni-mainz.de), which is a member of the AHRP and the Gauss Alliance e.V. The authors gratefully acknowledge the computing time granted on the supercomputer Mogon at Johannes Gutenberg University Mainz (www.hpc.uni-mainz.de).

The author wishes to acknowledge use of the PARAVIEW program by the Sandia National Laboratories. Sandia is a multiprogram laboratory operated by Sandia Corporation, a Lockheed Martin Company, for the United States Department of Energy's National Nuclear Security Administration under contract DE-AC04-94AL85000.

The author wishes to acknowledge use of the Ferret program for analysis and graphics in this paper. Ferret is a product of NOAA's Pacific Marine Environmental Laboratory; further information is available at http://ferret.pmel.noaa.gov/Ferret/.

Furthermore, the author thanks all MESSy developers and users for support; further information can be found at http://www.messy-interface.org.

The author thanks the two anonymous reviewers for their constructive comments, which helped to improve this manuscript.

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

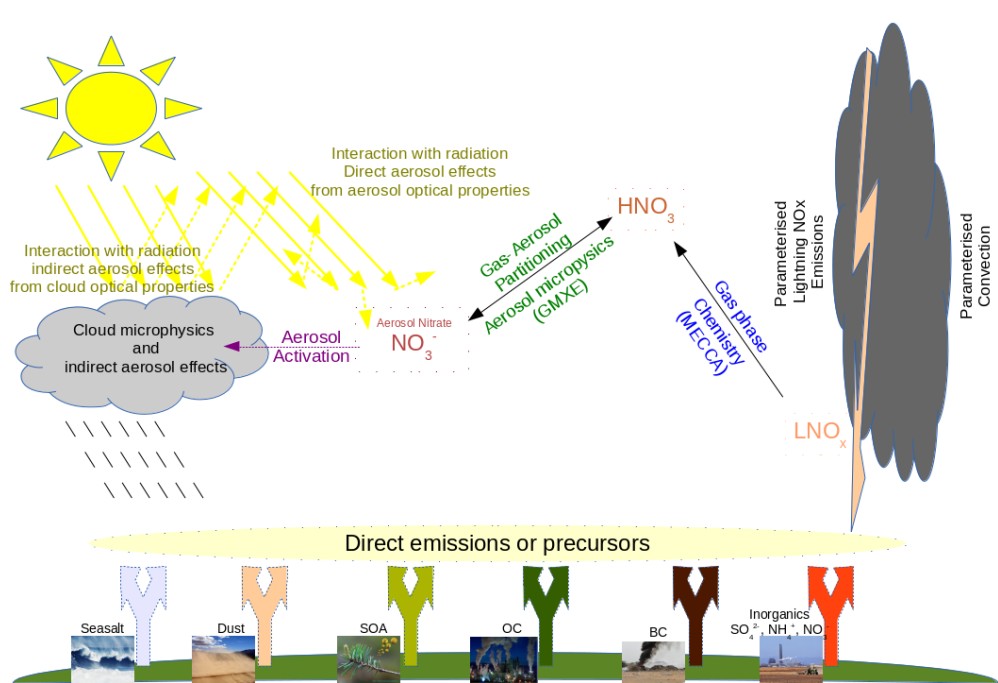

**Figure 1.** Sketch of the simulated processes from the emission of $NO_x$ molecules by lightning over gas phase conversion to $HNO_3$, gas aerosol partitioning to particulate $NO_3^-$ and implications for the direct and indirect aerosol effects.

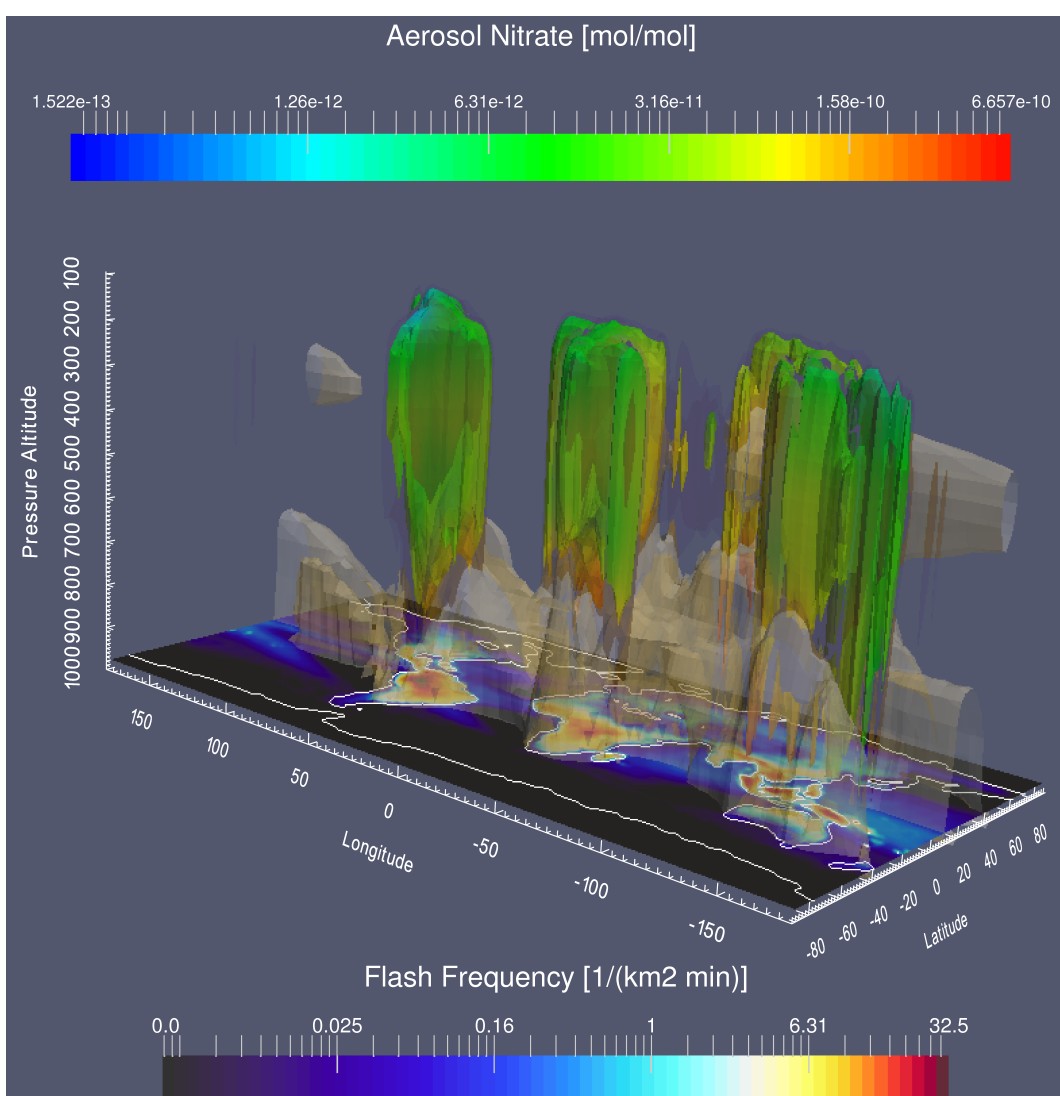

**Figure 2.** 3D visualisation of $LNO_x$ emissions (coloured isosurface of $1 \cdot 10^{-16}$ kg/(m$^3$s) and darker shaded isosurface of $3 \cdot 10^{-16}$ kg/(m$^3$s)) and the total aerosol nitrate mixing ratios (gray isosurface of 0.1 ppb$_v$). Additionally, the mean flash rate in 1/(km$^2$ min) is depicted by the 2D slice at the bottom. Note the logarithmic scaling of both colour bars.

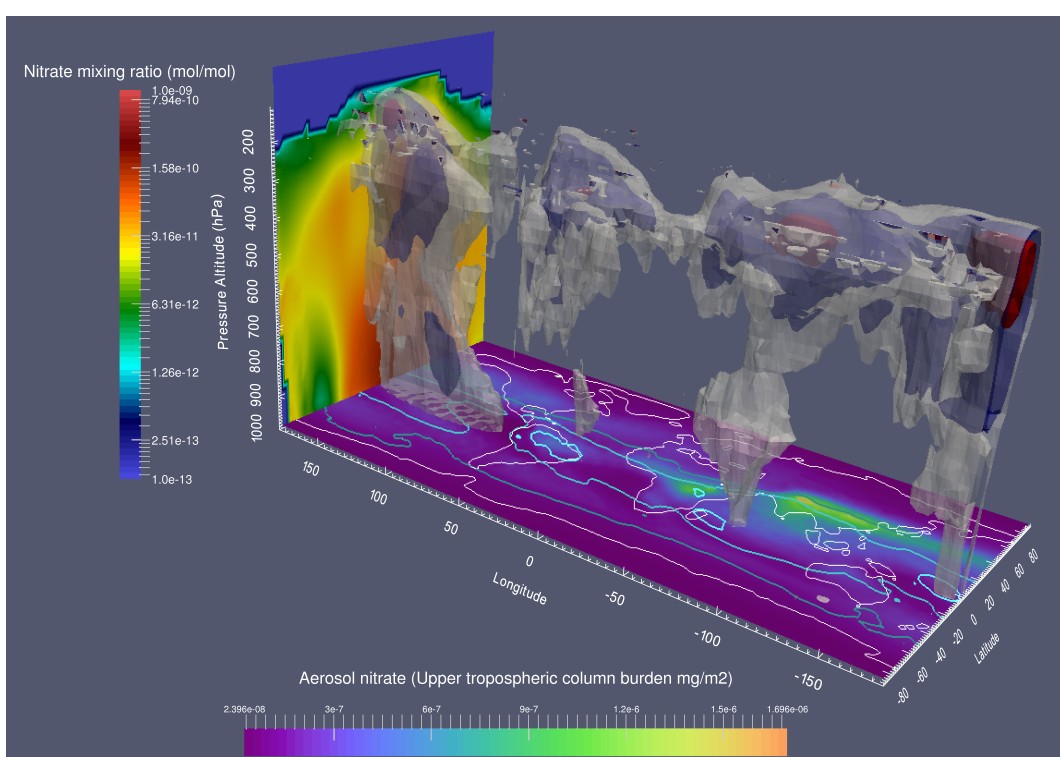

**Figure 3.** 3D visualisation of the relative differences in tropospheric aerosol nitrate mixing ratios between the simulations with and without $LNO_x$ emissions to the simulation including $LNO_x$ emissions. The gray isosurface depicts a relative difference of 30%, the blue isosurface of 45%, and the red isosurface of 60%. Additionally, the upper tropospheric aerosol nitrate column burden (in mg/m$^2$) between 500 hPa and the tropopause is depicted by the coloured panel at the bottom of the graph. The turquoise contour lines depict relative differences of 20%, 40% and 60% difference in this column burden between the two simulations.

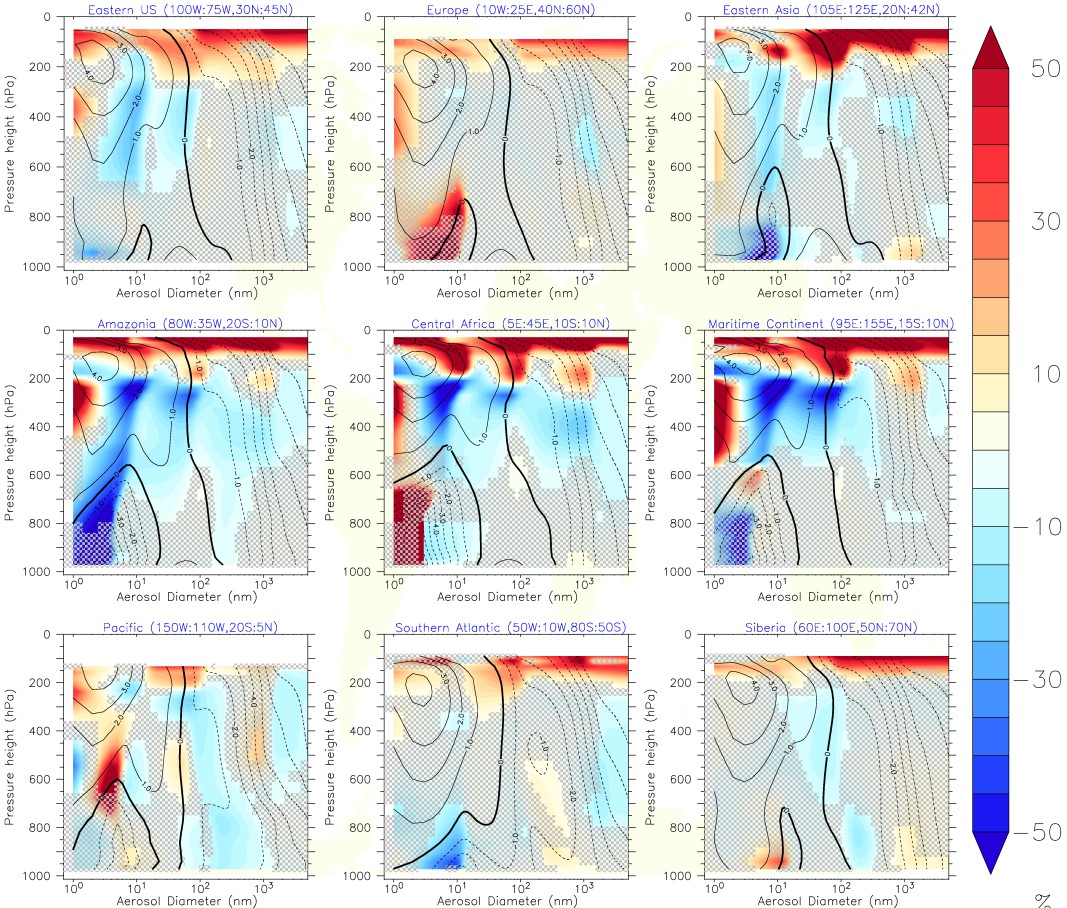

**Figure 4.** 9 panel plot of the vertically resolved percentage differences (including $LNO_x$ emissions as the reference case, i.e. (NOLNOX-LNOX)/LNOX) in the aerosol size distributions as spatial and regional average (for the respective regions). In each small frame the y-axis depicts pressure altitude and the x-axis the aerosol diameter from 1nm to 10 $\mu$m. Overlayed are the contours of the absolute values of the size distributions, i.e. the absolute particle numbers calculated from the overlaying of the individual modes, determined from the spatial and temporal mean in particles/cm$^3$. The solid lines depict 1, 10, 100, 1000 particles, whereas the dashed lines represent $10^{-1}, 10^{-2}$, etc. particles. The figure depicts the present day conditions. The respective regions are: Central Pacific (150W:110W, 20S:0), Amazonia (80W:35W, 20S:10N), Eastern US (100W:75W, 30N:45N), South Atlantic (50W:10W, 80S:50S), Central Africa (5E:45E, 10S:10N), Europe (10W:25E, 40N:60N), Indonesia (95E:155E, 12S:10N), East Asia (105E:125E, 20N:42N), Siberia (60E:100E, 50N:70N). The hatched regions mask areas without statistical significance.

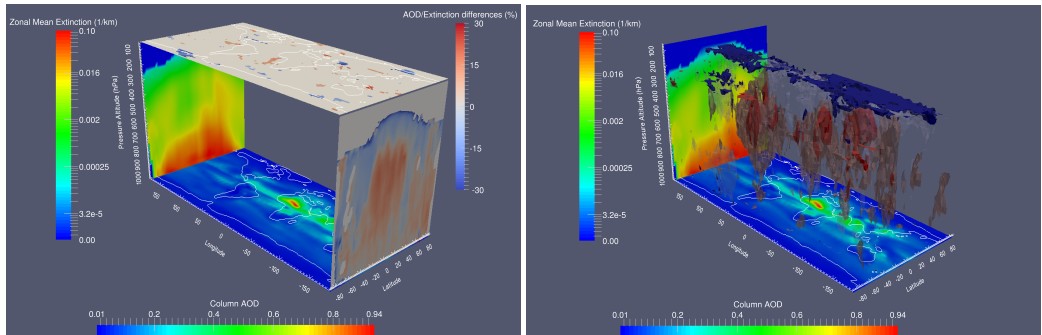

**Figure 5.** 3D visualisation of aerosol extinction and the influence of $LNO_x$ emissions. The left panel shows a map of the vertically integrated column AOD (at 550 nm) when lightning emissions are included (floor plane). The ceiling depicts the relative differences of the integrated column AOD between the simulation with lightning emissions minus the simulation without lightning $LNO_x$, with the full setup serving as reference. The back panel displays the zonal average aerosol extinction (in 1/km at 550 nm) of the full simulations (Please, note the logarithmic scale.). The front panel depicts relative percentage differences due $NO_x$ emissions from lightning. White areas mark regions without statistical significance.

The right panel also shows AOD (floor) and zonal mean extinction (rear panel), and additionally 3D isosurfaces in the center of the box represent the $+10\%$ (pale red) and $+20\%$ (dark red) of the enhanced extinction due to active $LNO_x$ emissions, whereas the $-10\%$ (pale blue) and $-20\%$ (dark blue) isosurfaces mark regions, in which the emissions result in a reduction of the extinction.

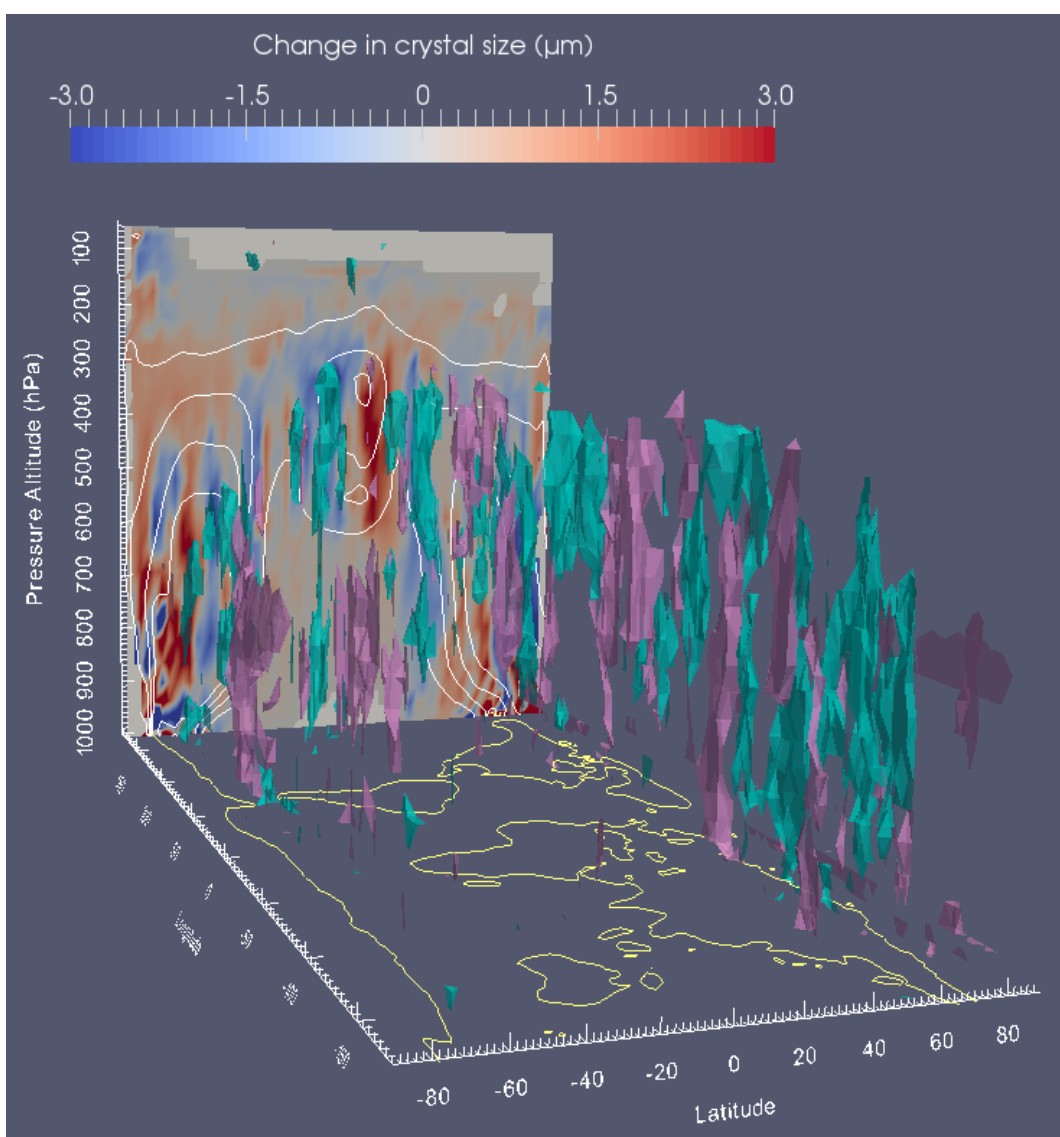

**Figure 6.** Visualisation of changes in the effective radius of ice crystals. The back panel displays the statistically significant absolute changes in zonal average ice crystal size in $\mu$m. The white contours depict a mean ice crystal size and therefore mark regions with large and smaller crystals. Additionally, the isosurfaces represent the regions for substantial absolute changes for the effective ice crystal size (turquoise for negative and purple for positive) due the $LNO_x$ emissions.

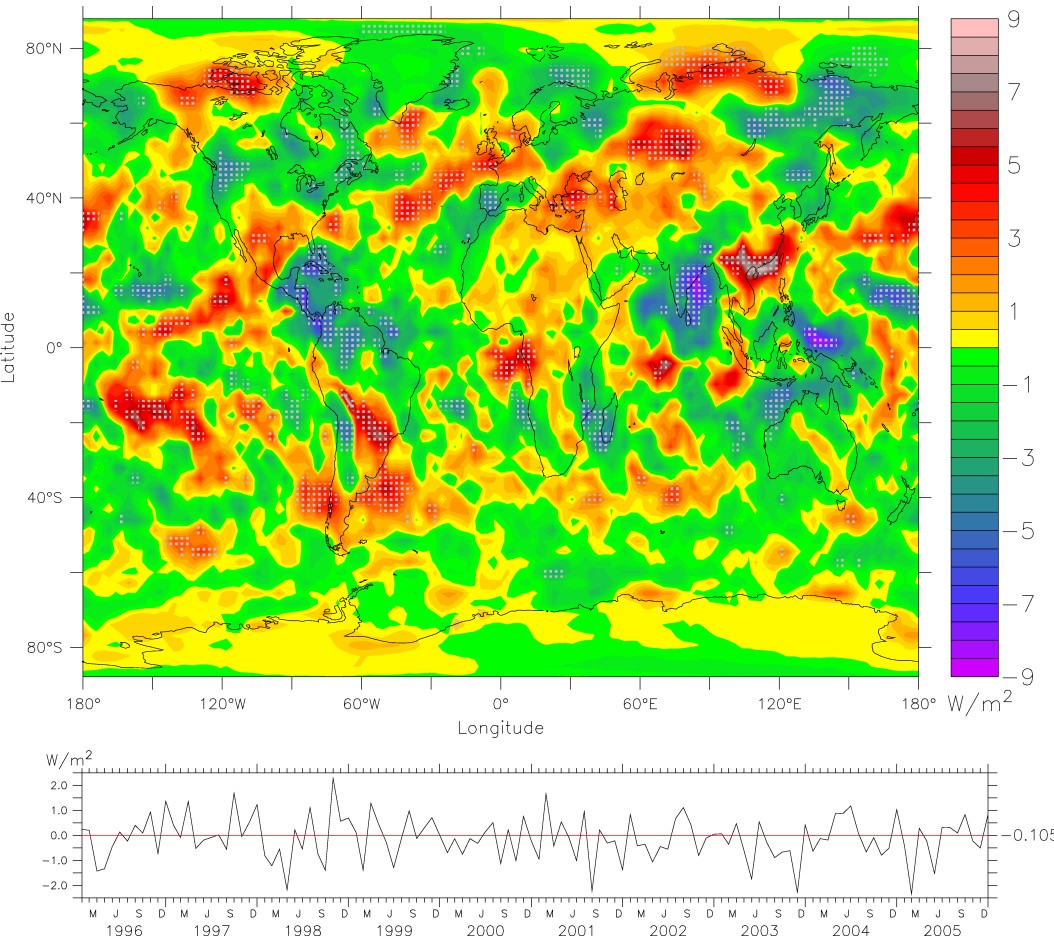

**Figure 7.** Absolute changes in the shortwave all-sky fluxes at the top of the atmosphere. The upper panel depicts the long term time average of the difference in the simulation with $LNO_x$ emissions minus the fluxes without lightning emissions. The hatches mark regions with a statistical significant signal (compared to the internal interannual variability). The bottom panel depicts the time series of the monthly mean global mean differences with the red line marking a 0 W/m$^2$ change.

**Table 1.** Tropospheric and upper tropospheric burden in the ARG simulation of the important highly oxidised nitrogen species, i.e. gaseous $HNO_3$, gaseous $N_2O_5$, aerosol $NO_3^-$ and the sum of those three compounds. All values are given in Gg N (except for the relative differences which are provided in %) and are globally and vertically integrated over the whole and the upper troposphere (500hPa up to the tropopause).

| | $HNO_3$ | $N_2O_5$ | $NO_3^-$ | Total N(V+) |
|---|---|---|---|---|
| **Present day** | | | | |
| Absolute values (with $LNO_x$ emissions): | | | | |
| Tropospheric Column burden | 403 | 8.0 | 143 | 554 |
| UT Column burden | 174 | 5.0 | 27.8 | 207 |
| Absolute differences due to $LNO_x$ emissions: | | | | |
| Tropospheric Column burden | 162 | 3.9 | 20.0 | 186 |
| UT Column burden | 107 | 3.5 | 9.0 | 119 |
| Relative differences in (%) due to $LNO_x$ emissions: | | | | |
| Tropospheric Column burden | 40.1 | 48.4 | 14.0 | 33.5 |
| UT Column burden | 61.3 | 70.3 | 32.3 | 57.7 |
| **Preindustrial conditions** | | | | |
| Absolute values (with $LNO_x$ emissions): | | | | |
| Tropospheric Column burden | 238 | 4.5 | 67.8 | 310 |
| UT Column burden | 143 | 4.1 | 15.0 | 161 |
| Absolute differences due to $LNO_x$ emissions: | | | | |
| Tropospheric Column burden | 158 | 3.7 | 25.8 | 188 |
| UT Column burden | 110 | 3.5 | 8.1 | 121 |
| Relative differences in (%) due to $LNO_x$ emissions: | | | | |
| Tropospheric Column burden | 66.6 | 82.6 | 38.1 | 60.6 |
| UT Column burden | 76.9 | 86.0 | 54.3 | 75.1 |

**Table 2.** $CH_4$ tropospheric and upper tropospheric (500 hPa to tropopause) lifetime and absolute change due to $LNO_x$ emissions.

| | $CH_4$ lifetime [years] | Increase of the $CH_4$ lifetime [years] |
|---|---|---|
| Present day | | |
| troposphere | 7.4 | 1.7 |
| UT only | 14.0 | 9.7 |
| Preindustrial conditions | | |
| troposphere | 8.9 | 1.9 |
| UT only | 14.1 | 13.2 |

The $CH_4$ lifetime is calculated with the help of the actual methane and OH concentrations, calculating a pseudo-first order loss rate. The corresponding temperature dependent reaction rate is identical to the one used in the online chemistry calculations, using also the temperature output of the simulations. Calculating the lifetime for each individual cell, average values are determined using grid mass weighted factors for certain regions or the whole troposphere.