# Peer review of "Chemistry-climate interactions of aerosol nitrate from lightning"

_Atmospheric Chemistry and Physics, 2016_

## Referee Comment (RC1) · Anonymous Referee #1 · 24 Aug 2016

Review of "Chemistry-climate interactions of aerosol nitrate from lightning" by Tost

Summary

This article shows the effect of nitrogen oxides ($NO_x$) produced by lightning on aerosol nitrate, other trace gases and aerosols, aerosol and cloud properties, and atmospheric shortwave radiation using a global chemistry-climate model. The impact of including lightning-$NO_x$ ($LNO_x$) emissions is to increase ozone and odd hydrogen (as shown via the methane lifetime), similar to previous studies. $LNO_x$ emissions increase aerosol nitrate burdens, and also aerosol sulfate burdens (somewhat) because of the effect on odd hydrogen. These changes manifest themselves in changes in aerosol size distributions and extinction, cloud drop and ice number concentrations, as well as a small effect on the radiation fluxes. The new information from this paper is the impact on aerosols. However, I am not convinced these are significant impacts because of the large internal variability and uncertainties in the modeling system.

The paper should focus on what is unique and on what the important findings are. In its current state, the paper- does not do a sufficient job of characterizing what changes are important and which ones are not. There is much refinement to the analysis on what these results mean that needs to be done. The presentation needs improvement too. I personally would not use 3-d figures, because they are difficult to understand. The English writing could be improved too.

Major Points

1. There are many discussion points illustrating small differences between simulations with and without $LNO_x$ emissions. Are these differences significant enough to be discussed? I suggest the author focus the paper on substantial (i.e. >10%) changes and what unique things we learn from these simulations. Because the effect of $LNO_x$ emissions on ozone and other oxidants has been shown before (e.g., Labrador et al., 2005), these findings should be limited to explaining why the sulfate aerosol burden increases. I think using actual oxidant burdens is preferable to using methane lifetime.

2. The paper contains discussion on the impact of $LNO_x$ emissions during preindustrial times, showing results in the supplement, but these results are similar to the present day scenario and do not even warrant mentioning in the abstract. Perhaps the preindustrial results could simply be summarized in a paragraph in the discussion or conclusions of this paper.

3. It seems to me that the loss processes should be addressed in this study. The author mentions the formation of $NH_4NO_3$ in the upper troposphere. However, considering inorganic aerosol particles are mostly washed out (e.g., Chatterjee et al., 2010, J. Atmos. Chem.; Gilardoni et al., 2014, ACP; Yang et al., 2015, JGR) and the highly soluble $NH_3$ and $HNO_3$ are likely removed by thunderstorms, it seems that there may not be enough $NH_3$ to form $NH_4NO_3$ downwind of the storms.

4. I thought many of the figures were challenging to read. The author should consider whether the figures do the best job in delivering the message of the paper

in a clear manner such that the readers can easily grasp the science learned. The author should also consider how well a colleague could explain the science in the paper using the figures provided (e.g. for teaching purposes).

Specific Comments

Lines 30-37. Is there any observational evidence of $NH_4NO_3$ formation downwind of convection?

Lines 107-118. Are aerosol-cloud interactions applied to both resolved and parameterized clouds? Please clarify.

Lines 134-136. It would be good to give a brief description of what ACCMIP present-day and pre-industrial emissions are. For example, what assumptions went into creating the pre-industrial scenario.

Lines 140-143. Are there specific years that comprise the present-day simulations? Are there specific years for pre-industrial simulations? Are all these simulations run as a climate model or are they driven from a reanalysis product?

Lines 151-152. Subsequent papers by Pickering's students have updated the vertical profiles for lightning-$NO_x$ emissions (DeCaria et al., 2005; Ott et al., 2010). These profiles exclude a NO source at lower altitudes. It seems that the consequence of the Pickering et al. (1998) profiles do not have a big effect on the model results, but I suggest that the chemistry-climate model be updated.

Lines 176-178. It seems that PAN and other organic nitrates and perhaps $NO_3$ should be included as contributors to $NO_y$ in addition to $NO_x$, $HNO_3$, and $N_2O_5$.

Lines 198-199. How long does it take for $NO_2$ to transform to aerosol nitrate?

Lines 200-203. It appears that a discussion of the results of Figure 3 was not included.

Lines 211-217. To me, a budget includes source and sink terms describing the major pathways creating and destroying a trace gas. Table 1 shows only the contribution of various species to N(V), and has no discussion of the processes affecting N(V) species.

Lines 215-217. It would be nice to see supporting information that $NH_3$ emissions are responsible for lower particulate nitrate concentrations.

Lines 211-217. It seems that $NO_3$, PAN, and other organic nitrates should be included.

Lines 218-223. I could not connect the numbers presented in this paragraph to the ones listed in the Table. Is the particulate nitrate contribution determined from dividing nitrate column burden by total N(V) column burden (143/554)?

Lines 232-235. Another recent paper also reports the effect of $LNO_x$ emissions on tropospheric ozone burden. Finney et al. (2016) ACP find a 27-30% increase in tropospheric ozone due to $LNO_x$ depending on the manner for calculating lightning

flash rates. The paper tropospheric ozone burden found by Finney et al. (2016) is substantially lower than that reported here. I assume these differences are simply based on the model configuration. However, it would be good to cite the Finney et al. (2016) findings.

The author may want to also cite Finney et al. (2016) GRL, which discusses the effect of $LNO_x$ emissions on ozone among the ACCMIP models.

Lines 245-251. As mentioned in the opening remarks in this review, the methane lifetime is not really the best way to show that OH is affected by $LNO_x$ emissions and therefore sulfate burdens. I suggest removing this discussion. However, if it is kept there are a few things that need to be improved.

First, please explain the method better for calculating the methane lifetime. Second, be consistent with nomenclature. In the text it says $LNO_x$ emissions increase methane lifetime, but in the table caption it says, "increase due to neglect of $LNO_x$ emissions". Third, please explain why the change in lifetime occurs. This might be discussed in Labrador et al. (2005), but it is worth summarizing in this paper.

Lines 253-257. It is interesting to see that the $LNO_x$ emissions affect sulfate aerosol concentrations. The author attributes this to the gas-phase chemical production via OH oxidation of sulfur dioxide. However, could the aqueous-phase production also be different because its main oxidants, ozone and hydrogen peroxide, are affected by $LNO_x$?

Section 3.2. The author highlights changes of various key constituents. Is it important to highlight the change if it is less than a 10% change? Surely, there is enough uncertainty in other model parameters to complicate the interpretation of a small change in the burden of a constituent. Perhaps the author could state the statistical significance of these changes.

Lines 279-290. It would be helpful to know why the 9 regions were chosen. It appears that the regions are defined by latitude-longitude values without regard for land or ocean (which have quite different aerosol size distributions). If the data were further filtered for over land regions for the U.S., South America, Africa, Europe, East Asia, and Siberia, would there be a substantial change in size distributions?

Lines 312-315. The impact of $LNO_x$ on aerosol water uptake is not surprising since most of the $LNO_x$ effect is in the upper troposphere where it is quite dry.

Lines 342-344. It is an interesting point that the maximum enhancement in aerosol extinction occurs in the middle troposphere. I think the strength of using a global model is to show these downstream effects. Can the author say something as to why the middle troposphere is affected more than the upper troposphere? I was going to suggest the ice sedimentation to the mid-troposphere where $HNO_3$ would be degassed when the ice sublimated, but the author points out that the largest aerosol nitrate enhancement is in the upper troposphere.

Lines 371-374. I like frequency distributions because they quantify some more the changes that are occurring. It seems to me that these plots could be included as a

figure, especially since it is worth discussing. The discussion seems to point to one perhaps significant difference between present day and pre-industrial scenarios.

Lines 401-402. Why is the polar latitude cloud coverage changed and is it *statistically* significant?

Lines 415-421. The discussion focuses on the cloud drop number and ice crystal number concentrations, but there should also be a few remarks about CCN and IN number concentration changes as well.

Technical Comments

L. 27. $LNO_x$ needs to be defined. I suggest doing that on Line 23.

L. 53. Insert "a" before "few".

L. 80. "NOx" needs a subscript "x" to be consistent with manuscript.

L. 84. Should it be "emitted NO" or "emitted $NO_x$"?

L. 121. "4" needs to be subscripted.

L. 153-159. This paragraph does not seem to belong in section 3.1 on lightning and $LNO_x$ emissions.

L. 161. Remove "also". I suggest a good proofreading to remove unnecessary "also"s and improve the writing in general.

L. 174. Replace "mixes with" with "along".

L. 189. Insert "of" between "factor" and "two".

L. 190. Move "globally averaged" to after "32%".

L. 192. The figure caption says it is a white isosurface, but on this line it says gray. Be consistent.

L. 200. Remove "Additionally".

L. 276. What are the values of the contour lines?

L. 277. Capitalize "for" at the end of the line. That is, start a new sentence.

L. 310. Remove "-" and use a comma.

L. 311. Remove "e.g."

L. 311. Remove "-" and use a comma.

Lines 322-333. Most of this paragraph discusses the effect of $LNO_x$ emissions on AOD. Is it intended to discuss aerosol extinction in this paragraph too? It is confusing, plus the sentences should be placed after discussing global aerosol extinction changes.

L. 336. Add "of aerosol extinction" after "relative change".

L. 339. Move "also" to just before "simulated".

L. 342. Add "of aerosol extinction" after "enhancement".

L. 350. "lighting" should be "lightning".

L. 367. Is this sentence discussing tropical South America or all of South America?

L. 369. It seems unnecessary to have both "extra-tropics" and "mid-latitude".

L. 401. Add a comma after "this".

L. 404. Insert "by" before "local" and insert "there are" after "but".

L. 410. Is the increase of 10% for total water content?

L. 413. Insert "ice" before "crystal".

L. 415. "Further North" of where?

L. 424. It is better to say "model output" rather than "data".

L. 428. Insert "By" before "analyzing". Remove "with the help of" and put parentheses around "Fig. 6", adding a comma after the parentheses.

L. 502. Remove "also".

L. 503. Change "load" to "loading".

L. 516. I suggest using "unclear" instead of "ambiguous".

L. 558. Why is a 2014 paper (Chang et al., ACP) a discussion paper? Please update!

Figure 2. The flashrate units do not match between the color bar on the plot and the figure caption. If the units were flashes per km$^2$ per minute, then it would be easier to compare to satellite data in the literature.

Figure 4. I suggest changing the color bar to have white for the -2% to +2% region. The -5 to -15% colors are so similar it is difficult to see changes. The same is true with -30 to -40% and 18-32%. For such small plots, perhaps it is better to have just 5 colors: red, yellow, white, green, blue, and define broader regions of percent difference.

Figure 4 caption. What are the contour level values?

Figure 5. "Additionally the front panel depicts again relative percentage differences" of what? And no need for both "additionally" and "again".

Figure 6. It should be "effective radius". What are the values of the isosurfaces? "substantial absolute changes" is not quantitative.

Figure 6. Why are there ice crystal size changes in the 1000-700 hPa regions where it is usually too warm to support ice?

---

## Referee Comment (RC2) · Anonymous Referee #2 · 10 Sep 2016

**1 Summary**

This study performs a series of sensitivity simulations in a global coupled chemistry-climate model (CCM) aimed at characterizing the impact of lightning $NO_x$ emissions on the composition of the atmosphere and climate, with particular focus on the role of nitrate aerosol. Within the EMAC CCM, the study has performed decadal time-slice simulations for the preindustrial and present. For each time period, simulations with zero lightning emissions were compared to simulations containing lightning emissions. All simulations were done with two separate mechanism for cloud-aerosol interactions.

The paper is of scientific merit and the results interesting. However, I think that in order to be suitable for publication, there needs to be a careful look at the statistical

significance of the results over such a short simulation period (see general comments). I'm not convinced that the study period is of appropriate length for the system being examined.

**2  General Comments**

1. I'm concerned that 10-year time-slice simulations are too short a period over which to significantly quantify lightning-aerosol impacts in a free-running chemistry-climate model. Both lightning and aerosols have strong sensitivity to clouds, which are highly variable in space and time. Whereas the global chemical tendencies and their physical explanations as argued here are probably correct, I'm not sure how well we can trust the reported magnitudes without an analysis of how statistically significant the changes are relative to the natural climate variability over the period. The weak pattern correlation reported in Section 3.4.1 and the large variability relative to the lightning $NO_x$ forcing observed in the latter figures imply that there is still low signal-to-noise. The figure that does show significance tests (Fig. 7), has few locations with statistical significance, which are generally not regions with largest impacts from lightning. I'm especially concerned about the cloud properties changes attributed to lightning in the difference plots. I think the manuscript would be improved if the same significance tests were done for the data in Figs. 2-6 to establish which signals are robust. Ideally, the simulations could be extended until significance was achieved in each of the examined variables. However, I realize that this is not necessarily possible, so I think at least including the statistical significance estimates are critical, if the simulated changes are to be attributed to the lightning emissions.

2. The author is correct that few global CCMs include aerosol nitrate. However, the paper neglects to mention the global and regional chemical transport models

(CTMs) studies of lightning impacts on photochemistry and aerosols, several of which include ammonium nitrate aerosol thermodynamics and chemistry. Some recent examples include

- Allen et al., ACP, 2012, doi:10.5194/acp-12-1737-2012
- Murray et al., JGR, 2013, doi:10.1002/jgrd.50857
- Holmes et al., ACP, 2013, doi:10.5194/acp-13-285-2013
- Zare et al., ACP, 2014, doi:10.5194/acp-14-2735-2014
- Gressant et al., ACP, 2016, doi:10.5194/acp-16-5867-2016

Whereas, none of those studies explicitly report the impact of lightning on nitrate aerosol, the influence of lightning via nitrate aerosol pathways on ozone, OH, methane lifetime, and/or bulk $PM_{2.5}$ were included. In particular, the conclusion stated on page 8, lines 241-244 implies that no lightning photochemistry study has included nitrate aerosol chemistry. I would recommend that the Introduction and Section 3.2.2 be rephrased to acknowledge that the ozone, OH and methane lifetime results presented here are in agreement with the CTM studies, and emphasize that what is uniquely reported here are (1) the isolation of lightning impacts on nitrate aerosol, and (2) the discussion of lightning impacts on climate-relevant aerosol properties and the climate system itself.

3. The 3-D renders in the figures are novel and interesting, but very hard to interpret in a 2-D print media, especially for Fig. 5 and 6, where information on the faces of the rectangular cube are severely distorted and blocked by the 3-D contours. In particular, I think Fig. 5 would benefit by being converted into multiple figures.

**3   Specific Comments**

- P2; L47 - "this" indirect effect

- P2; L53-63 - This study is still a substantially large perturbation to the reactive nitrogen budget of the free troposphere, so I would still consider its results to be strongly susceptible to errors introduced by non-linearities. To truly minimize these errors, one would need to do a small perturbation analysis (e.g., Sauvage et al., 2007; doi:10.1029/2006JD008008). I agree with the method used here, but I think this paragraph is slightly misleading with respect to the uncertainties.

- Section 2.3 - How many years was each simulation initialized over? Is methane prescribed or allowed to respond to the large changes in OH?

- P3; L82 - comma should be before "we" instead of "that"

- P5; L132 - nitrate "precursors" from

- P5; L152 - The "C-shaped" profiles are somewhat outdated. Unimodal distributions with maxima in the free troposphere suggested by top-down (Ott et al., 2010; doi:10.1029/2009JD01188) and bottom-up modeling studies (Koshak et al., 2014; doi:10.1016/j.atmosres.2012.12.015).

- P5; L161 - "also" should be moved to before "are"

- P8, L251 - do you mean "oxidation capacity of the upper troposphere"?

- P9; L289-290 - I'm not sure exactly what is meant by the clause that begins with "whereas." But if there is a statistically significant increase in CCN over Africa (but not South American or Indonesia), that is an interesting result from the perspective of the potential role that it might play in leading to convective invigoration that might contribute to the African lightning maximum, which models seldom replicate. Aerosols have been implicated before (e.g., Jacobson et al., 2009; doi:10.1029/2008JD01147).

- P9; L334 - I interpret "back" and "front" panels as being the northern and southern faces of the rectangular cube. I would recommend switching to using "left" and "right" face, or "western" and "eastern" face for Figs. 5 and 6.

- P14; L465-466 - Lightning strongly impacts background global oxidant levels. Shouldn't we expect significant impacts on shortwave radiation near the aerosol precursor sources in the midlatitudes?

- P15; L497 - comma should go after "NH3"

- P15; L499 - please clarify if you mean to the global aerosol nitrate burden, or the local upper troposphere

- P15; L504 - Why not represent the oxidation potential via the oxidant concentrations themselves, rather than indirectly via the lifetime? The methane lifetime is heavily biased toward the tropics due to strong temperature sensitivity of the $CH_4$ + OH reaction. This may underrepresent the importance of lightning on the extratropics.

- Fig. 4 - the axis labels for the contour panels are illegible. I would recommend making this a 3 x 3 panel plot, with the panels in rough geographic order.

- Fig. 7 - the units for the y-axis are missing (W m$^{-2}$?)

- Table 1 - "Differences due to LNO$_x$ emissions" is ambiguous toward its directionality. I'd recommend using "Estimated contribution from lightning emissions".
* * *

---

## Author Comment (AC1) · 5 Dec 2016

**Reply to reviewer 1 of the**
**interactive comment on "Chemistry-climate interactions of aerosol nitrate from lightning"**

I thank the reviewer for his comments which helped to improve the manuscript.
The reviewer comments are given in italics and my comments are following the individual points.

In the summary, the reviewer requests are stronger focus on the new findings on aerosols. However, as some changes in aerosols and their properties can only be derived from the impact of LNOx on the gas phase these effects have to be mentioned as well. I agree that some of the discussion on gas phase chemistry can be shortened as this has already been the focus on numerous studies.
A major concern is the statistical significance of the results which is addressed in the revised manuscript version. Many of the impacts on aerosols and their properties show are statistically robust despite the model internal and interannual variability.
I agree that the model formulation include a lot of simplifications which lead to model uncertainties, but nevertheless the model is a state-of-the-art chemistry climate model and this study is a first attempt to analyse and quantify the impact of lightning on nitrate aerosols.
The 3D visualisations, even though a bit more challenging for understanding at first glance allow the inclusion of more information in a single graph, i.e. the exact geographical location of isolated three dimensional features which cannot be depicted by 2D cross sections or 2D averages. The updated manuscript therefore still contains 3D graphs, but 2D sections are provided in the supplement which may be more convenient to visualise individual aspects. Additionally, replacing the 3D visualisations would result in substantially more pictures in the manuscript increasing its length and decreasing the readability.

**Major points:**
1)
*There are many discussion points illustrating small differences between simulations with and without LNOx emissions. Are these differences significant enough to be discussed? I suggest the author focus the paper on substantial (i.e. >10%) changes and what unique things we learn from these simulations. Because the effect of LNOx emissions on ozone and other oxidants has been shown before (e.g., Labrador et al., 2005), these findings should be limited to explaining why the sulfate aerosol burden increases. I think using actual oxidant burdens is preferable to using methane lifetime.*

The revised manuscript depicts only statistically significant differences. For nitrate almost all upper tropospheric differences show statistical robustness. Using derived aerosol properties and their impacts on the climate system the robustness becomes smaller, especially for liquid water clouds the statistical significance is not reached. Therefore this part is substantially shortened in the revised version.
The isosurfaces in the 3D visualisations always show changes larger than 10%, sometimes even up to 60%, and also the impact on CH4 lifetime for the whole troposphere is larger than 20% (which is also statistically robust). Consequently, we can follow the process chain as depicted in Fig.1 from the NOx source from lightning up to radiative flux perturbations as a result from neglecting LNOx emissions.
I agree that most of the gas phase results have been published before (and the revised manuscript will even include some more references), but few studies allow to draw conclusion about the implications of the gas phase on the aerosol phase. Furthermore, I think that the fact that considering aerosol particles does not substantially change the results from pure gas phase studies is also an important point.

2)
*The paper contains discussion on the impact of LNOx emissions during preindustrial times, showing results in the supplement, but these results are similar to the present day scenario and do not even*

*warrant mentioning in the abstract. Perhaps the preindustrial results could simply be summarized in a paragraph in the discussion or conclusions of this paper.*

I agree that the results for preindustrial conditions do not provide any new insights and a mostly suitable to determine the internal variability thus providing additional confidence in the robustness of the simulation results. Furthermore, as LNOx emission contribute more to total NOx emissions under preindustrial conditions the effects of lightning are actually even stronger in this scenario.
In the revised version the preindustrial scenario will be discussed in an appendix, and not in every section of the manuscript.

3)
*It seems to me that the loss processes should be addressed in this study. The author mentions the formation of NH4NO3 in the upper troposphere. However, considering inorganic aerosol particles are mostly washed out (e.g., Chatterjee et al., 2010, J. Atmos. Chem.; Gilardoni et al., 2014, ACP; Yang et al., 2015, JGR) and the highly soluble NH3 and HNO3 are likely removed by thunderstorms, it seems that there may not be enough NH3 to form NH4NO3 downwind of the storms.*

Even though nitrate particles formed near the surface will be mostly removed by wet deposition, similar to HNO3 the formation of NH4NO3 in the upper troposphere is still a real phenomenon. C-ToF-AMS Measurements during ACRIDICON-CHUVA (a campaign in Brazil with the HALO aircraft) taking place in the convective outflow in the upper troposphere report quite often enhanced nitrate values which are highly consistent with enhanced NOx and NOy data. (pers. Communication, J. Schneider, MPI for Chemistry Mainz, Nov. 2016). The stabilising cations in this altitude are either ammonium or potentially potassium from biomass burning. However, the latter source is not included in the presented model simulations, such that NH4+ ions are the only choice for neutralising cations. Furthermore, Höpfner et al., (ACP, 2016) detected NH3 in the gas phase in the outflow of the Asian summer monsoon, which then could also form the NH4+ ions. Consequently, the wet removal of NH3 might not be complete such that sufficient ammonia and subsequent ammonium can be available for the formation of NH4NO3.

4)
*I thought many of the figures were challenging to read. The author should consider whether the figures do the best job in delivering the message of the paper in a clear manner such that the readers can easily grasp the science learned. The author should also consider how well a colleague could explain the science in the paper using the figures provided (e.g. for teaching purposes)*

As mentioned above I prefer one picture using 3D visualisation over several figures using 2D graphs, since this allows a more consistent representation of the real distributions. I agree that some graphs are too busy, especially Fig.5 and 6 which I have refurbished for the revised manuscript version. Additionally, some more graphs in 2D have been added to the supplement, which are easier to read and therefore potentially for suitable for teaching purposes.
Nevertheless, I personally think that 3D visualisation offers a potential for data analysis which has often been neglected in the past because of higher level of complexity. Usually the authors know the data they plot very well, however, a reader of a manuscript who has no access to the data often is not provided with all information the authors have available to come to their conclusions. This is (in my opinion) slightly improved when 3D visualisation is used.

**Specific comments:**

*Lines 30-37. Is there any observational evidence of NH4NO3 formation downwind of convection?*
As mentioned above, observational evidence exists for nitrate from aerosol mass spectrometry, however only the anion has been detected. But as gaseous NH3 is also found in this altitude, the existence for NH4NO3 is very likely.

*Lines 107-118. Are aerosol-cloud interactions applied to both resolved and parameterized clouds? Please clarify.*
The current convection scheme in the EMAC model system is not aware of aerosol particles (except for vertical transport). Therefore aerosol cloud interactions are only considered for large-scale clouds and detrained cloud water/ice to form cirrus clouds in the anvil regions of convective activity. This is clarified in the revised manuscript version.

*Lines 134-136. It would be good to give a brief description of what ACCMIP present-day and pre-industrial emissions are. For example, what assumptions went into creating the pre-industrial scenario.*
The answers to this question is quite extensive. For the current simulation this is of minor importance (in my point of view), but more details on the ACCMIP simulations and the respective emissions can be found in the overview paper of Lamarque et al. (GMD, 2013).

*Lines 140-143. Are there specific years that comprise the present-day simulations? Are there specific years for pre-industrial simulations? Are all these simulations run as a climate model or are they driven from a reanalysis product?*
The simulations have been conducted as free running simulations with a climatological SST and sea-ice coverage (AMIP2 climatology). The simulation has been initialised with data for the year 1996, but afterwards the simulation has not been nudged towards a re-analysis product. Therefore, the processes and feedback are not weakened or amplified by an external data source for neither present nor preindustrial conditions. Chemical initialisation has been taken from the same year from a transient simulation within the ESCiMo project (Jöckel et al., GMD, 2016). The emissions have been kept constant for the whole simulation time, but of course include a seasonal cycle.

*Lines 151-152. Subsequent papers by Pickering's students have updated the vertical profiles for lightning-NOx emissions (DeCaria et al., 2005; Ott et al., 2010). These profiles exclude a NO source at lower altitudes. It seems that the consequence of the Pickering et al. (1998) profiles do not have a big effect on the model results, but I suggest that the chemistry-climate model be updated.*
I am aware of the updated LNOx emission profiles, and currently the new distributions are implemented as alternative vertical emission distribution in the modelling system. However, from previous studies with our model system (e.g. Huntrieser et al., 2007 for the TROCCINOX campaing and Tost et al., 2010 for SCOUT-O3-DARWIN) lead to the conclusion that the Pickering vertical distribution scheme in combination with the applied convection and convective transport algorithm result in simulation results which agree well with aircraft observations.

*Lines 176-178. It seems that PAN and other organic nitrates and perhaps NO3 should be included as contributors to NOy in addition to NOx, HNO3, and N2O5.*
Indeed PAN is very important for long-range transport of NOy, but for the sake of limiting myself to the aerosol phase as much as possible, PAN is not considered in the analysis. Nevertheless, it is part of the chemical reaction system and therefore included in the model simulations. Higher organonitrates are currently lumped together and also contribute to the NOy reservoirs. The total PAN load of the atmosphere decreases by approx. 15% in case that LNOx emissions are neglected. This corresponds

almost perfectly to the reduction in nitrates. However, the upper troposphere reduction of PAN is with 21% substantially lower compared to the nitrate reduction with 32%. This is a consequence of the composition for PAN which does not only require NOy, but also VOC compounds. Consequently, in case LNOx emissions are taken into account, maximum PAN formation is not always limited by the amount of available NOy, but the limitation is caused by the low amount of VOCs in the upper troposphere.

*Lines 198-199. How long does it take for NO2 to transform to aerosol nitrate?*
The formation time is typically in the order of several hours to a few days, and therefore one of the main conversion term for NOx into NOy, especially in the upper troposphere.

*Lines 200-203. It appears that a discussion of the results of Figure 3 was not included.*
The discussion of Fig.3 is extended in the revised version.

*Lines 211-217. To me, a budget includes source and sink terms describing the major pathways creating and destroying a trace gas. Table 1 shows only the contribution of various species to N(V), and has no discussion of the processes affecting N(V) species.*
The loss processes for the N(V) species are tabulated in the supplement to close the budget.

*Lines 215-217. It would be nice to see supporting information that NH3 emissions are responsible for lower particulate nitrate concentrations.*
*Lines 211-217. It seems that NO3, PAN, and other organic nitrates should be included.*
The analysis of the corresponding simulation data has led to the conclusions in the manuscript (without analysing PAN in detail). Again, focussing on the aerosol and not the full N(V) budget has led to these shortenings in the manuscript, which I would like to keep this way. The response of PAN has been discussed shortly above.

*Lines 218-223. I could not connect the numbers presented in this paragraph to the ones listed in the Table. Is the particulate nitrate contribution determined from dividing nitrate column burden by total N(V) column burden (143/554)?*
Indeed, this way the numbers in this text section are determined.

*Lines 232-235. Another recent paper also reports the effect of LNOx emissions on tropospheric ozone burden. Finney et al. (2016) ACP find a 27-30% increase in tropospheric ozone due to LNOx depending on the manner for calculating lightning flash rates. The paper tropospheric ozone burden found by Finney et al. (2016) is substantially lower than that reported here. I assume these differences are simply based on the model configuration. However, it would be good to cite the Finney et al. (2016) findings.*
I agree that the results of (Finney et al., ACP, 2016) are interesting and related to the findings in my study. The increase in the tropospheric burden in my study is around 225 for the whole troposphere; this is in agreement with Labrador et al., 2005. For the upper troposphere the increase is more important with 27% which is comparable to the results from Finney for the whole troposphere. Depending on the model configuration, especially the vertical resolution, the transport of ozone from the stratosphere into the upper troposphere can have substantial impact on the tropospheric ozone burden. However, the effect of total NOx emissions (from all sources) as well as total VOC emissions can have a similar effect on the tropospheric ozone burden, e.g. an increase in isoprene emissions due to a slightly different leaf area index can lead to a substantial increase in the tropospheric ozone burden. Consequently, the model configuration as well as the interplay of various boundary conditions can cause the difference in the total tropospheric O3 burden. In total the EMAC model tends to have a

slight high bias in tropospheric ozone (c.f. Jöckel et al. 2016).

*The author may want to also cite Finney et al. (2016) GRL, which discusses the effect of LNOx emissions on ozone among the ACCMIP models.*
This study is added in the introduction.

*Lines 245-251. As mentioned in the opening remarks in this review, the methane lifetime is not really the best way to show that OH is affected by LNOx emissions and therefore sulfate burdens. I suggest removing this discussion. However, if it is kept there are a few things that need to be improved. First, please explain the method better for calculating the methane lifetime. Second, be consistent with nomenclature. In the text it says LNOx emissions increase methane lifetime, but in the table caption it says, "increase due to neglect of LNOx emissions". Third, please explain why the change in lifetime occurs. This might be discussed in Labrador et al. (2005), but it is worth summarizing in this paper.*
I tend to disagree in this point, as oxidant concentrations fail to consider the recycling potential of oxidants, especially OH. The OH burden of the atmosphere differs only by 10% between the simulations with lightning compared to the simulation without LNOx emissions, whereas the methane lifetime effect is twice that large. For calculating the $CH_4$ lifetime, the $CH_4$ and OH concentrations as well as temperature and the second order reaction rate which is also used in the chemistry mechanism of the EMAC model.
The table caption is changed according to the reviewers suggestion.
The $CH_4$ lifetime is mostly affected by the change in the $O_3$ mixing ratios, which directly influence the amount of OH produced from O3P (from photolysed $O_3$) and $H_2O$. Furthermore, the OH recycling reaction $NO + HO_2 \rightarrow NO_2 + OH$ also directly depends on the available $NO_x$ concentration. The impact of the second formation (or better recycling reaction) cannot be directly analysed with considering the OH concentration only (or the $HO_x$ concentrations).

*Lines 253-257. It is interesting to see that the LNOx emissions affect sulfate aerosol concentrations. The author attributes this to the gas-phase chemical production via OH oxidation of sulfur dioxide. However, could the aqueous-phase production also be different because its main oxidants, ozone and hydrogen peroxide, are affected by LNOx?*
As the effects of lightning are strongest in the upper troposphere, but the aqueous phase production is most efficient in the lower troposphere due to the amount of available liquid water (as shown e.g. by Tost et al., ICCP, 2012), the aqueous phase production is less affected by the LNOx emissions. Since less S(VI) is formed via the gas phase, even more S(IV) is available for aqueous phase oxidation. These two effects almost balance each other, such that there are only insignificant effects on aqueous phase sulfate formation.

*Section 3.2. The author highlights changes of various key constituents. Is it important to highlight the change if it is less than a 10% change? Surely, there is enough uncertainty in other model parameters to complicate the interpretation of a small change in the burden of a constituent. Perhaps the author could state the statistical significance of these changes.*
The only change that is lower than 10% is the change in the sulfate burden. Even though this change is small and in some locations not statistically significant, it is in agreement with the significant changes in the size distribution. Therefore, I came to the conclusion that the reduced sulfate production is of relevance and decided to mention it in the manuscript.

*Lines 279-290. It would be helpful to know why the 9 regions were chosen. It appears that the regions are defined by latitude-longitude values without regard for land or ocean (which have quite different aerosol size distributions). If the data were further filtered for over land regions for the U.S., South*

*America, Africa, Europe, East Asia, and Siberia, would there be a substantial change in size distributions?*

The regions have been selected based on the degree of lightning activity and to a second degree on the changes in nitrate concentrations. Even though continental and oceanic boxes are combined usually the continental boxes dominate the size distribution due to the substantially higher aerosol burden, such that the distributions are representative for continental conditions. Only the marine regions in the Central Pacific and South Atlantic are representative for oceanic conditions. In the revised version the size distribution changes are also checked for statistical significance. This reveals that only fine mode differences in the middle and upper troposphere exhibit robustness.

*Lines 312-315. The impact of LNOx on aerosol water uptake is not surprising since most of the LNOx effect is in the upper troposphere where it is quite dry.*

I agree and the sentence will be rephrased to explicitly mention this point. Nevertheless, I also checked changes in hygroscopicity and also this revealed very little effects. As AOD is mostly affected by aerosol water, I found it important to analyse whether aerosol water had an impact on both the size distribution as well as the extinction coefficients.

*Lines 342-344. It is an interesting point that the maximum enhancement in aerosol extinction occurs in the middle troposphere. I think the strength of using a global model is to show these downstream effects. Can the author say something as to why the middle troposphere is affected more than the upper troposphere? I was going to suggest the ice sedimentation to the mid-troposphere where HNO3 would be degassed when the ice sublimated, but the author points out that the largest aerosol nitrate enhancement is in the upper troposphere.*

I think that nitrate sedimentation plays a minor role here, since this would be represented already in the concentration differences. As stated below, the pattern correlation of the change in extinction corresponds much better to the change in the sulfate concentrations. The change in the extinction led to the analysis of the size distributions and revealed the connection to the sulfate changes.

*Lines 371-374. I like frequency distributions because they quantify some more the changes that are occurring. It seems to me that these plots could be included as a figure, especially since it is worth discussing. The discussion seems to point to one perhaps significant difference between present day and pre-industrial scenarios.*

The frequency distributions help to quantify the differences; however, in this case the plots exhibit only small differences that have not been tested for significance. In the PDFs all data points and not the mean values have been used. I found the results sufficiently interesting to mention, but they have not been conclusive to differentiate cases under present day and preindustrial conditions.

*Lines 401-402. Why is the polar latitude cloud coverage changed and is it statistically significant?*

The polar cloud coverage change is not statistically significant, even though it is larger than in most other regions. The wording in the manuscript has been misleading.

*Lines 415-421. The discussion focuses on the cloud drop number and ice crystal number concentrations, but there should also be a few remarks about CCN and IN number concentration changes as well.*

Changes in CCN and IN can partly be deduced from the aerosol size distributions. As liquid clouds hardly show significant differences, the CCN distributions are most likely also not significant. Concerning the IN, these are not explicitly diagnosed by the model, but crystal formation is parameterised based on several aerosol species. The shape of the regions of significant ice crystal size changes leads to the conclusion that immersion freezing in mixed phase clouds is the dominant process

responsible for the differences.

**Technical Comments:**
*L. 27. LNOx needs to be defined. I suggest doing that on Line 23.*
Agreed.

*L. 53. Insert "a" before "few".*
done

*L. 80. "NOx" needs a subscript "x" to be consistent with manuscript.*
checked throughout the manuscript.

*L. 84. Should it be "emitted NO" or "emitted NOx"?*
LNOx emissions are completely put into the tracer mixing ratio of NO.

*L. 121. "4" needs to be subscripted.*
done

*L. 153-159. This paragraph does not seem to belong in section 3.1 on lightning and LNOx emissions.*
The paragraph will be rephrased. It should explain the grey isosurface of elevated nitrate concentrations in the lower troposphere, as well as the nitrate mixing ratios in the LNOx emission plumes.

*L. 161. Remove "also". I suggest a good proofreading to remove unnecessary "also"s and improve the writing in general.*
The manuscript is going to be checked before re-submission. Before final publication the mandatory copy-editing by copernicus will also help to improve the language of a non-native speaker.

*L. 174. Replace "mixes with" with "along".*
done
*L. 189. Insert "of" between "factor" and "two".*
done
*L. 190. Move "globally averaged" to after "32%".*
done
*L. 192. The figure caption says it is a white isosurface, but on this line it says gray. Be consistent.*
corrected
*L. 200. Remove "Additionally".*
done
*L. 276. What are the values of the contour lines?*
The contour lines represent absolute number concentrations, with the thick line representing 1 particle, dashed lines $10^{-1}$, $10^{-2}$, etc. and continuous lines $10^1$, $10^2$ etc. particles per $cm^{-4}$. Zooming into the plot the actual values become visible.

*L. 277. Capitalize "for" at the end of the line. That is, start a new sentence.*
done
*L. 310. Remove "-" and use a comma.*
*L. 311. Remove "e.g."*
*L. 311. Remove "-" and use a comma.*
done

*Lines 322-333. Most of this paragraph discusses the effect of LNOx emissions on AOD. Is it intended to discuss aerosol extinction in this paragraph too? It is confusing, plus the sentences should be placed after discussing global aerosol extinction changes.*
This paragraph is rephrased taking the statistical significance into account. First AOD and afterwards extinction are discussed.

*L. 336. Add "of aerosol extinction" after "relative change".*
done
*L. 339. Move "also" to just before "simulated".*
done
*L. 342. Add "of aerosol extinction" after "enhancement".*
done
*L. 350. "lighting" should be "lightning".*
done
*L. 367. Is this sentence discussing tropical South America or all of South America?*
Tropcial South America has been meant.

*L. 369. It seems unnecessary to have both "extra-tropics" and "mid-latitude".*
*L. 401. Add a comma after "this".*
*L. 404. Insert "by" before "local" and insert "there are" after "but".*
done

*L. 410. Is the increase of 10% for total water content?*
yes

*L. 413. Insert "ice" before "crystal".*
done
*L. 415. "Further North" of where?*
Transition from tropics to mid-latitudes.

*L. 424. It is better to say "model output" rather than "data".*
*L. 428. Insert "By" before "analyzing". Remove "with the help of" and put parentheses around "Fig. 6", adding a comma after the parentheses.*
*L. 502. Remove "also".*
*L. 503. Change "load" to "loading".*
done

*L. 516. I suggest using "unclear" instead of "ambiguous".*

*L. 558. Why is a 2014 paper (Chang et al., ACP) a discussion paper? Please update!*
A final publication of the Change et al., ACPD manuscript is not foreseen. A new manuscript describing the approach is currently under review for Atmospheric Environment.

*Figure 2. The flashrate units do not match between the color bar on the plot and the figure caption. If the units were flashes per km2 per minute, then it would be easier to compare to satellite data in the literature.*
Caption corrected.
*Figure 4. I suggest changing the color bar to have white for the -2% to +2% region. The -5 to -15% colors are so similar it is difficult to see changes. The same is true with -30 to -40% and 18-32%. For*

*such small plots, perhaps it is better to have just 5 colors: red, yellow, white, green, blue, and define broader regions of percent difference.*
*Figure 4 caption. What are the contour level values?*
Figure refurbished and caption corrected.

*Figure 5. "Additionally the front panel depicts again relative percentage differences" of what? And no need for both "additionally" and "again".*
Caption and figure updated.

*Figure 6. It should be "effective radius". What are the values of the isosurfaces?*
*"substantial absolute changes" is not quantitative.*
*Figure 6. Why are there ice crystal size changes in the 1000-700 hPa regions where*
*it is usually too warm to support ice?*
Caption and figure updated; most of the changes for liquid clouds have not been significant, as well as ice crystal size changes in non-ice regions.

---

## Author Comment (AC2) · 5 Dec 2016

**Reply to reviewer 2 of the**
**interactive comment on "Chemistry-climate interactions of aerosol nitrate from lightning"**

I thank the reviewer for his comments which helped to improve the manuscript.
The reviewer comments are given in italics and my comments are following the individual points.

Major comments:
1)
*I'm concerned that 10-year time-slice simulations are too short a period over which to significantly quantify lightning-aerosol impacts in a free-running chemistry-climate model. Both lightning and aerosols have strong sensitivity to clouds, which are highly variable in space and time. Whereas the global chemical tendencies and their physical explanations as argued here are probably correct, I'm not sure how well we can trust the reported magnitudes without an analysis of how statistically significant the changes are relative to the natural climate variability over the period. The weak pattern correlation reported in Section 3.4.1 and the large variability relative to the lightning NOx forcing observed in the latter figures imply that there is still low signal-to-noise. The figure that does show significance tests (Fig. 7), has few locations with statistical significance, which are generally not regions with largest impacts from lightning. I'm especially concerned about the cloud properties changes attributed to lightning in the difference plots. I think the manuscript would be improved if the same significance tests were done for the data in Figs. 2-6 to establish which signals are robust. Ideally, the simulations could be extended until significance was achieved in each of the examined variables. However, I realize that this is not necessarily possible, so I think at least including the statistical significance estimates are critical, if the simulated changes are to be attributed to the lightning emissions.*

I agree with the reviewer that some of the signals do not appear to be robust. However, I have recalculated the data used in the graphs and in the revised version of the manuscript only the statistical significant changes are displayed. Nevertheless, the changes in aerosol nitrate from lightning (Fig.3) are very robust, such that the significance test shows that they are statistical significant and the graph does not substantially change. Similarly the main features in the changes of the size distributions (Fig.4) are also statistically significant. The updated version of the manuscripts marks all non-significant areas of the plot with hashes. The conclusions drawn from the previous version of the graph remain unchanged since they are focused on the large differences which exhibit statistical significance. Analysing the changes in extinction with respect to statistical significance also reveals that the influence of the lightning emissions (both the additional nitrate formation as well as the sulphate production) are robust compared to the internal interannual variability, such that the influence on extinction remains visible. However, analysing the statistical significance of the differences in column AOD reveals that these changes are mostly not significant. This is also not very surprising since the differences in extinction are located in the upper troposphere, but this region does only contribute to the total AOD to a minor degree (see back panel in Fig.5a), as the extinction rates are a factor of 10 or even 100 smaller compared to the near surface values which dominate the column AOD.
Concerning the changes in effective radius these are hardly significant for cloud droplets, but the statistical significance for the ice crystals is larger. Consequently, the impact of the cloud droplet activation scheme is of minor importance as warm clouds play a secondary role in changes due to lightning emissions. Nevertheless, clouds (also warm clouds)  contribute to the statistical noise in the radiative fluxes analysed in Fig.7 such that the regions with significant changes in the radiative fluxes cannot unambiguously attributed to the lightning emissions any more. The updated version of the figure including a correction of a small mistake in the significance test also shows slightly more statistical significance especially in the regions of substantial cooling. Despite the problem with a direct

co-location of sources and effects, the global total effect is robustly negative in all simulations, both for present day and pre-industrial conditions with both warm cloud activation schemes, such that a total cooling effect can in my opinion be determined from the simulations.

I personally have doubts that extending the simulations will substantially improve statistical robustness of the results. Even though mathematically the number of data points included in the significance test scales with the power of 0.5 to the significance and the variability is supposed to not increase, the cloud effects, which still have the highest level of uncertainty with respect to process understanding, substantially contribute to the total radiative effect and this conclusion can already be drawn from the current simulation length. For a continuation of the simulation time the computing resources have not been available, such that the answer here is only speculative.

2)

*The author is correct that few global CCMs include aerosol nitrate. However, the paper neglects to mention the global and regional chemical transport models (CTMs) studies of lightning impacts on photochemistry and aerosols, several of which include ammonium nitrate aerosol thermodynamics and chemistry. Some recent examples include:*

*Allen et al., ACP, 2012, doi:10.5194/acp-12-1737-2012*
*Murray et al., JGR, 2013, doi:10.1002/jgrd.50857*
*Holmes et al., ACP, 2013, doi:10.5194/acp-13-285-2013*
*Zare et al., ACP, 2014, doi:10.5194/acp-14-2735-2014*
*Gressant et al., ACP, 2016, doi:10.5194/acp-16-5867-2016*

*Whereas, none of those studies explicitly report the impact of lightning on nitrate aerosol, the influence of lightning via nitrate aerosol pathways on ozone, OH, methane lifetime, and/or bulk PM2.5 were included. In particular, the conclusion stated on page 8, lines 241-244 implies that no lightning photochemistry study has included nitrate aerosol chemistry. I would recommend that the Introduction and Section 3.2.2 be rephrased to acknowledge that the ozone, OH and methane lifetime results presented here are in agreement with the CTM studies, and emphasize that what is uniquely reported here are (1) the isolation of lightning impacts on nitrate aerosol, and (2) the discussion of lightning impacts on climate-relevant aerosol properties and the climate system itself.*

Again I agree, that the impact of lightning on nitrate aerosol, especially the chemical composition the oxidation capacity of the atmosphere and PM have been previously investigated. As correctly stated by the reviewer, none of these simulations included the complex feedback on the dynamics of the atmosphere (as they have been conducted with transport models). I admit that I have not been aware of all these simulations, and I will mention some of them in the introduction. However, several of these studies have been conducted with regional models such that a global perspective cannot immediately be offered. Furthermore, some studies also neglect the particulate phase. I have shown that the impact of the particulate phase on the gas phase is minor, however the inversion of this statement is not the valid, e.g. the Allen et al. Study analyses the impact of lightning on nitrate deposition, but does not take the consequences for sulphate explicitly into account.

In the revised introduction, the complex interactions in this study allowing the multi-directional feedback between lightning, gas phase chemistry, particulate phase and the impact on the dynamics of the atmosphere via radiation and cloud processes will be better elucidated to show the novelties of the current study.

3)

*The 3-D renders in the figures are novel and interesting, but very hard to interpret in a 2-D print media, especially for Fig. 5 and 6, where information on the faces of the rectangular cube are severely distorted and blocked by the 3-D contours. In particular, I think Fig. 5 would benefit by being*

*converted into multiple figures.*

I think that the 3D visualisation offers the benefits of displaying in more detail regions of interest. I agree that some of the graphs are more difficult to understand at first glance in contrast to some 2D visualisations, but they offer the potential to include more information in the same number of graphs. Otherwise, a substantial increase in the number of figures would be required to show all the conclusion drawn in the analysis, which is all included in the individual 3D visualisations. Fig.5 and Fig.6 are revised due to the results based on the statistical significance (see above) such that some of the information has been removed from the figures allowing a better visibility of the main features. Additionally Fig.5 has been replaced by two graphs, one showing the main statistical significant regions in combined two dimensional structures and the other one depicting the 3D structure of the impact of lighting on extinction.

Specific comments:
• *P2; L47 - "this" indirect effect*
  corrected
• *P2; L53-63 - This study is still a substantially large perturbation to the reactive nitrogen budget of the free troposphere, so I would still consider its results to be strongly susceptible to errors introduced by non-linearities. To truly minimize these errors, one would need to do a small perturbation analysis (e.g., Sauvage et al., 2007; doi:10.1029/2006JD008008). I agree with the method used here, but I think this paragraph is slightly misleading with respect to the uncertainties.*

This study is not such a strong annihilation scenario comparing simulations with nitrate to those without any nitrate; nevertheless it is still an annihilation scenario with respect to LNOx emissions.

The formulation is rephrased. I agree that the disturbance is still large due to the impact on the chemical regimes and hence the oxidation capacity of the atmosphere. A disturbance study would be better suited, e.g. 2 Tg LNOx emissions, 5 Tg LNOx emissions and 8 Tg LNOx emissions. However, the computation time for this study (which already encompassed 80 years of simulation time with a comprehensive chemistry climate model including gas and cloud phase chemistry as well as aerosol particles) has been limited such that these sensitivity studies could not have been conducted. Furthermore, for these cases most likely the signal-to-noise ratio would have been even worse such that no conclusion might have been drawn from these simulation results.

• *Section 2.3 - How many years was each simulation initialized over? Is methane prescribed or allowed to respond to the large changes in OH?*

The simulation has been initialised with data from a comprehensive transient simulation (Jöckel et al., GMD, 2016) such that no additional spin-up phase has been conducted. Methane is prescribed at the surface with observed concentrations such that the change in the loss rates is partially dampened by additional pseudo-emissions. Therefore, the direct changes in the oxidants have not been reported, but the impact on the CH4 lifetime as a measure of the oxidation capacity as this quantity is less dependent on the actual methane concentrations.

• *P3; L82 - comma should be before "we" instead of "that"*
corrected

• *P5; L132 - nitrate "precursors" from*
reformulated

• *P5; L152 - The "C-shaped" profiles are somewhat outdated. Unimodal distributions with maxima in the free troposphere suggested by top-down (Ott et al., 2010; doi:10.1029/2009JD011880ls PDF*
*) and bottom-up modeling studies (Koshak et al., 2014; doi:10.1016/j.atmosres.2012.12.015).*

Even though unimodal distributions are found to give a better representation of the distribution of LNOx in present day studies, C-shaped profiles are found to give realistic results (at least in agreement with measurement campaigns (e.g. SCOUT-O3-Darwin (see Tost et al., 2010), TROCCINOX (Huntrieser et al., 2007)). As this is the current implementation of the vertical LNOx emission distribution function, this is not going to be changed. However, I will mention the Ott and Koshak studies.

• *P5; L161 - "also" should be moved to before "are"*
corrected

• *P8, L251 - do you mean "oxidation capacity of the upper troposphere"?*
The reduction to 50% corresponds to the upper troposphere only. The total oxidation capacity of the atmosphere is not affected this drastically.

• *P9; L289-290 - I'm not sure exactly what is meant by the clause that begins with "whereas." But if there is a statistically significant increase in CCN over Africa (but not South American or Indonesia), that is an interesting result from the perspective of the potential role that it might play in leading to convective invigoration that might contribute to the African lightning maximum, which models seldom replicate. Aerosols have been implicated before (e.g., Jacobson et al., 2009; doi:10.1029/2008JD01147).*
The changes in the size distribution in the lower part of the troposphere are found not to be statistically significant in neither South America nor Indonesia. Even though some significant changes are found in Central Africa, I do not see a direct link to a convective invigoration. On all three tropical continents the majority of the lower tropospheric aerosol particles result from biomass burning and SOA formation. The contribution of nitrates from lightning is small compared to the other sources in the lower troposphere and the feedback via the oxidation capacity and oxidative ageing of organic aerosols to increase their hygroscopicity and therefore cloud formation potential is not included in the model.

• *P9; L334 - I interpret "back" and "front" panels as being the northern and southern faces of the rectangular cube. I would recommend switching to using "left" and "right" face, or "western" and "eastern" face for Figs. 5 and 6.*
Fig.5 and 6 are revised. The left hand side of Fig.5 still uses the panels on the front and the back side of the cube – but I rather would not change the terminology into the geographic directions to avoid misunderstandings with regions on the globe.

• *P14; L465-466 - Lightning strongly impacts background global oxidant levels. Shouldn't we expect significant impacts on shortwave radiation near the aerosol precursor sources in the midlatitudes?*
Most of the oxidation of tropospheric aerosol precursors happens closer to the sources, which are located at the surface for most aerosol precursors ($SO_2$ from anthropogenic and $NO_x$ from anthropogenic and biogenic sources). In the lower part of the troposphere the oxidation capacity is not that substantially affected by lightning as in the upper troposphere. Especially sulphate formation is dominated by aqueous phase production, and the transfer of the oxidants into the aqueous phase is affected only to a minor degree.
I have analysed the sulphate production pathways in a different simulation scenario (without dynamical feedback) and have seen changes on the order of a few percent only.

• P15; L497 - comma should go after "NH3"
corrected

• *P15; L499 - please clarify if you mean to the global aerosol nitrate burden, or the local upper troposphere*

This statement is mostly valid for the upper troposphere and this is added in revised manuscript.

• *P15; L504 - Why not represent the oxidation potential via the oxidant concentrations themselves, rather than indirectly via the lifetime? The methane lifetime is heavily biased toward the tropics due to strong temperature sensitivity of the CH4 + OH reaction. This may underrepresent the importance of lightning on the extratropics.*

I prefer methane lifetime, since it is less dependent on the total CH4 concentrations. Furthermore, the recycling of oxidants especially OH via various reaction pathways cannot be well represented in oxidant concentrations. The difference in the OH burden is on the order of 10% only, whereas the CH4 lifetime for the troposphere has a magnitude of more than 20%, which results from the recycling potential. Consequently, CH4 lifetime is a better estimate for the oxidation capacity.

• *Fig. 4 - the axis labels for the contour panels are illegible. I would recommend making this a 3 x 3 panel plot, with the panels in rough geographic order.*

The axis labels are pressure altitude on the y-Axis and aerosol diameter on the x-Axis. The statistical significance has been added to the plots such that the important changes can be easier visualised. The Figure caption is changed to include this information as well.

• *Fig. 7 - the units for the y-axis are missing (W m$^{-2}$ ?)*

Both the color bar and the y-axis of the line plot depict the flux perturbation in W/m$^2$. This is mentioned below the color bar and at the upper edge of the y-axis.

• *Table 1 - "Differences due to LNOx emissions" is ambiguous toward its directionality. I'd recommend using "Estimated contribution from lightning emissions".*

As the difference can potentially be negative as well, I prefer the difference due to LNOx emissions. Furthermore, due to the complicated feedback, a contribution from lightning emissions might be misleading as the results can also be consequences of chemical feedback processes.

---

## Author Response (AR3)

Authors response file for
**Chemistry climate interactions of aerosol nitrate from lightning by H. Tost**

The reviewers comments have been taken into account.

A slight refurbishing of the text has been undertaken in a few spots, however, I did not have the impression that the flow of reading is interrupted by additions in case that the previous manuscript version is not known.

Figure 4 has been completely refurbished according to the reviewer's suggestions. The manuscript has been modified accordingly.

*1. Lines 42-47 contain the response to whether there is observational evidence of NH4NO3 formation in convective outflow regions. The response provides evidence of nitrate and NH3 separately. Along those lines, I would like to point the author to a recent paper presenting results on DC3 aerosol measurements:*

*Yang et al. (2015) Aerosol transport and wet scavenging in deep convective clouds: A case study and model evaluation using a multiple passive tracer analysis approach. J. Geophys. Res. Atmos., 120, 8448–8468, doi: 10.1002/2015JD023647.*

The study is added in the reference list, and an implication from this study has been added to the manuscript introduction.

"A recent study by Yang et al. (2015) also shows that the observed scavenging efficiencies for nitrate (together with $HNO_3$) and ammonium are around 80% for the DC3 campaign (Barth et al., 2015), which allows the conclusion that the observed $NO_3^-$ is most likely formed above the levels of substantial wet removal, e.g. by conversion of lightning $NO_x$ ."

*2. Lines 304-311. It would be good to include some of the remarks in the response to reviewer comments here. For example, the paragraph should state why the methane lifetime is used instead of OH itself:*

*"The OH burden of the atmosphere differs by only 10% …, but these differences do not reveal the impact on the oxidation capacity of the atmosphere that the methane lifetime reflects. The emissions of LNOx are responsible for …"*

The manuscript has been modified to take this statement into account.

"The effect on OH is displayed by a modification of the methane lifetime to additionally take the OH recycling capacity into account, as depicted in Tab. 2. The emissions of LNOx are responsible for an increase of the tropospheric methane lifetime of ∼ 1.7 to 1.9 years; however in the upper troposphere (above 500 hPa up to the tropopause) an increase of the CH4 lifetime of almost 10 and 13.2 years for present and preindustrial conditions, respectively, corresponding to almost a halving of the oxidation capacity of the upper troposphere. The direct OH concentrations differ only by ∼ 10 to 20% showing the importance of the recycling mechanisms on the oxidation capacity of the atmosphere."

*3. Lines 313-315. These lines state that the aqueous-phase production of sulfate is not important*

*because liquid water is in the lower troposphere. I do not find this to be a completely satisfying response. Are ozone and hydrogen peroxide mixing ratios significantly changed in the lower troposphere? If so, then aqueous phase sulfate production could be important. If not, then please revise to also say the oxidants do not change much either.*

The oxidants are affected to a lesser degree. This has been added to the manuscript.

"As most of the oxidation in the upper troposphere takes place in the gas phase, the aqueous phase oxidation in the lower troposphere is affected to a minor degree. Additionally, most of the emitted $SO_2$ in the lower troposphere originates from anthropogenic sources, where also the co-emission of NOx is prevalent such that the oxidant levels for aqueous phase oxidation of S(IV) to S(VI) are less affected."

*4. Lines 423-425. After reading the text again, I realize that the author's response to why the extinction maximum is in the middle of the troposphere is correct and stated in the subsequent paragraph of the manuscript. However, the text is not clearly written connecting lines 423-425 to the next paragraph. Perhaps a phrase can be added to line 425 that says "as will be shown next, sulfate aerosols are responsible for this enhancement in extinction". Or revise the first sentence of the subsequent paragraph to better connect the discussion.*

Text modified:
"...simulated between 200 and 400 hPa. The reason for this downward shift is discussed further below."

*5. Figure 4. I agree with the other reviewer in that Figure 4 can be read much better as a 9-panel plot without the map in background. Increase the size of the panels and have less empty space in the figure.*

*Figure 4. I would still like to see changes in the color bar so that it is easier to read. For me, it is especially important to change the colors with values near 0 to be white. I think the statistically significant regions will be more prominent by making this change.*

The Figure has been refurbished with an alternative colour scale and has been converted into a 9-panel plot, following the reviewers suggestions. Also a small error has been corrected for the South Atlantic region, which is of minor relevance for the conclusions. The text has been modified accordingly to these changes.

[revised manuscript text omitted]